# A genome-wide association analysis identifies 16 novel susceptibility loci for carpal tunnel syndrome

Akira Wiberg[1,2,3], Michael Ng[1], Annina B. Schmid[2], Robert W. Smillie[1], Georgios Baskozos [2], Michael V. Holmes[4,5], K. Künnapuu[6], R. Mägi[7], David L. Bennett [2] & Dominic Furniss [1,3]

Carpal tunnel syndrome (CTS) is a common and disabling condition of the hand caused by entrapment of the median nerve at the level of the wrist. It is the commonest entrapment neuropathy, with estimates of prevalence ranging between 5–10%. Here, we undertake a genome-wide association study (GWAS) of an entrapment neuropathy, using 12,312 CTS cases and 389,344 controls identified in UK Biobank. We discover 16 susceptibility loci for CTS with $p < 5 \times 10^{-8}$. We identify likely causal genes in the pathogenesis of CTS, including *ADAMTS17*, *ADAMTS10* and *EFEMP1*, and using RNA sequencing demonstrate expression of these genes in surgically resected tenosynovium from CTS patients. We perform Mendelian randomisation and demonstrate a causal relationship between short stature and higher risk of CTS. We suggest that variants within genes implicated in growth and extracellular matrix architecture contribute to the genetic predisposition to CTS by altering the environment through which the median nerve transits.

[1] Nuffield Department of Orthopaedics, Rheumatology, and Musculoskeletal Science, University of Oxford, Botnar Research Centre, Windmill Road, Oxford OX3 7LD, UK. [2] Nuffield Department of Clinical Neurosciences, University of Oxford, John Radcliffe Hospital, Oxford OX3 9DU, UK. [3] Department of Plastic and Reconstructive Surgery, Oxford University Hospitals NHS Foundation Trust, John Radcliffe Hospital, Oxford OX3 9DU, UK. [4] Medical Research Council Population Health Research Unit at the University of Oxford, Oxford OX3 7LF, UK. [5] Clinical Trial Service Unit & Epidemiological Studies Unit (CTSU), Nuffield Department of Population Health, University of Oxford, Richard Doll Building, Old Road Campus, Roosevelt Drive, Oxford OX3 7LF, UK. [6] Institute of Technology, University of Tartu, Nooruse 1, 50411 Tartu, Estonia. [7] Estonian Genome Center, Institute of Genomics, University of Tartu, Riia 23 B, 51010 Tartu, Estonia. Correspondence and requests for materials should be addressed to D.L.B. (email: david.bennett@ndcn.ox.ac.uk) or to D.F. (email: dominic.furniss@ndorms.ox.ac.uk)

Carpal tunnel syndrome (CTS) is a common and debilitating condition of the hand caused by the entrapment of the median nerve at the level of the wrist. It is the most common entrapment neuropathy, with estimates of prevalence ranging between 5 and 10%[1–3]. Symptoms include pain, paraesthesia, and numbness of the hand, and thenar weakness, leading ultimately to severe functional impairment[4]. Many patients require surgery to decompress the carpal tunnel, and although surgery is successful in the majority of patients, a significant subgroup experiences persistent or recurrent symptoms[5]. Thus, CTS exacts a considerable socioeconomic burden[6], and the number of CTS operations performed is projected to nearly double between 2011 and 2030[7].

Despite being such a common condition, the pathophysiology of CTS is poorly understood, and even less is known about the genetic contribution to the disease. Increased extraneural pressures have been strongly implicated in the pathophysiology of CTS[8]; this is believed to impair intraneural blood flow[9], leading to demyelination and eventual axonal loss[10]. CTS is also associated with fibrosis and thickening of the connective tissues that surround the median nerve and the flexor tendons within the carpal tunnel[11–13].

A large UK twin study reported a heritability of 0.46, suggesting that genetic factors are the strongest risk factors for CTS[14]. Furthermore, a positive family history is reported in 27–39% of cases of CTS[15]. A Swedish study calculated the standardised incidence ratio of CTS in siblings as 4.08, compared to a ratio of 2.06 among spouses, again suggesting a genetic influence on aetiology[16]. Environmental factors also play a role in CTS, and several occupational risk factors have been identified[17,18]. Moreover, CTS has a sex-specific incidence approximately four times higher in women than in men[19], and several systemic conditions are associated with CTS, including diabetes, obesity, rheumatoid arthritis, hypothyroidism and gout[20,21].

Thus, CTS can be considered a complex disease, in which genetic predisposition and environmental factors interact to affect the overall phenotypic expression. No genome-wide association study (GWAS) for CTS has been reported so far. With the aim of discovering genetic variants that confer risk to CTS, we undertook a GWAS for CTS in the UK Biobank resource, a prospective cohort study of ~500,000 individuals from the UK, aged between 40 and 69 years, who have had whole-genome genotyping undertaken and have allowed the linkage of these data with their medical records[22]. This is followed by gene-expression analyses in connective tissues collected from the carpal tunnels of the CTS patients to provide biological insight into those gene-association signals.

We found 16 genome-wide significant susceptibility loci for CTS and identified biologically plausible genes that could be causal in the pathogenesis of CTS. Using RNA sequencing, we demonstrated the expression of these genes in surgically resected tenosynovium from CTS patients. A Mendelian randomisation analysis revealed a causal relationship between short stature and higher risk of CTS. We suggest that variants within the genes implicated in skeletal growth and extracellular matrix architecture contribute to the genetic predisposition to CTS by altering the environment through which the median nerve transits.

## Results

**Association analysis.** Following sample- and single-nucleotide polymorphism (SNP)-based quality control (QC), we defined 12,312 participants of white British ancestry from this cohort with at least one diagnostic code for CTS as our cases (Supplementary Table 1) and used the remaining 389,344 white British participants as controls (Supplementary Fig. 1). Genome-wide association testing was undertaken across 547,011 common frequency-genotyped SNPs (minor allele frequency (MAF) $\geq 0.01$) and ~8.4 million imputed SNPs (MAF $\geq 0.01$, Info score $\geq 0.9$) using a linear mixed non-infinitesimal model implemented in BOLT-LMM v2.3[23] to account for population structure and relatedness. We assumed an additive genetic effect and conditioned on sex and the genotyping platform.

We discovered genome-wide significant associations ($p < 5 \times 10^{-8}$) at 422 variants across 16 loci (Fig. 1; Supplementary Data 1). The most significantly associated SNP at each locus is shown in Table 1. The $\lambda_{GC}$ demonstrated some inflation (1.15), but the linkage disequilibrium (LD) score (LDCS) regression intercept[24] of 1.015 with an attenuation ratio of 0.073 indicated that the inflation was largely due to polygenicity and the large sample size. Imputed SNPs were of high quality: the Info score was $\geq 0.924$ for the most significantly associated SNPs at each locus. We performed conditional analysis based on the top associated SNP at each locus and did not observe any additional independent association with CTS.

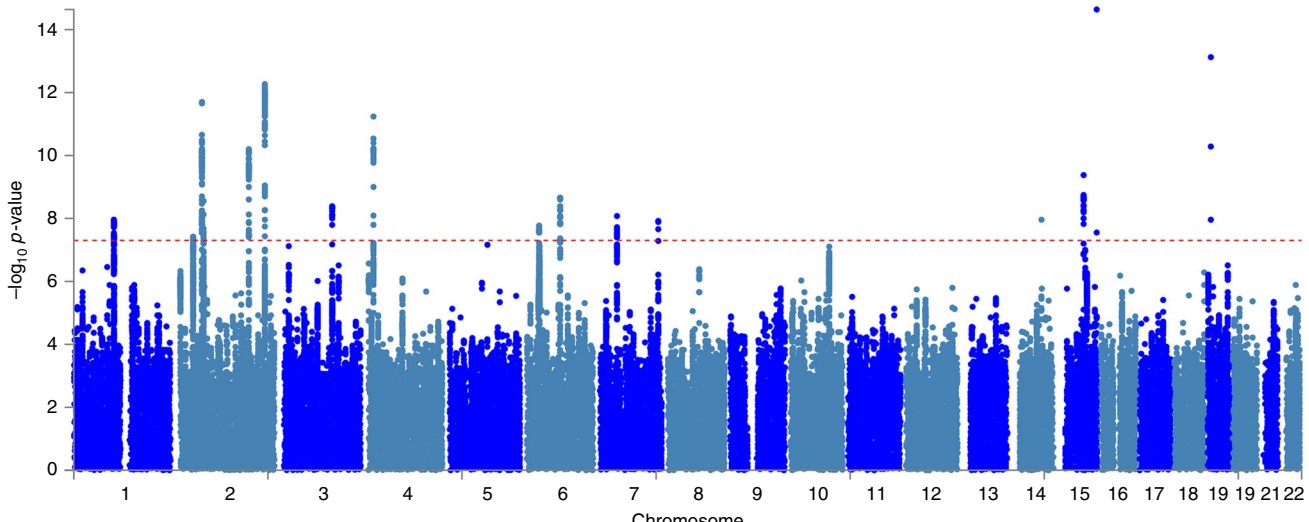

**Fig. 1** SNP associations with carpal tunnel syndrome. Manhattan plot showing -log$_{10}$ p values for SNP associations, produced in FUMA[25]. The horizontal red line represents $p = 5 \times 10^{-8}$. SNP, single-nucleotide polymorphism

**In silico analysis**. The two most significantly associated SNPs in the GWAS, rs72755233 (OR = 1.18, $p = 2.3 \times 10^{-15}$) and rs62621197 (OR = 1.31, $p = 7.5 \times 10^{-14}$), are missense variants in *ADAMTS17* and *ADAMTS10*, respectively (Fig. 2). In order to annotate and map the associated non-coding variants to known and predicted regulatory elements, we used FUMA[25]. Across

several of the loci there was strong evidence of functionality: 30 out of 422 genome-wide significant variants had a combined annotation-dependent depletion (CADD) score >12.37, the threshold suggested for deleterious SNPs[26]. This included the two missense variants above and rs3791679, the fourth most significantly associated SNP (OR = 1.11, $p = 2.0 \times 10^{-12}$), which

**Table 1 SNPs significantly associated with carpal tunnel syndrome**

| Chromosome | Position[a] | rsID | Effect allele | EAF cases[b] | EAF controls[c] | Info[d] | OR (95% CI) | p Value | Mapped genes[e] |
|---|---|---|---|---|---|---|---|---|---|
| 1 | 103240351 | rs12406439 | T | 0.605 | 0.587 | 0.995021 | 1.08 (1.05–1.11) | $1.10 \times 10^{-8}$ | COL11A1 |
| 2 | 33413303 | rs12104955 | C | 0.517 | 0.499 | 0.994032 | 1.07 (1.05–1.10) | $3.90 \times 10^{-8}$ | LTBP1 |
| 2 | 56096892 | rs3791679 | G | 0.244 | 0.225 | G | 1.11 (1.08–1.15) | $2.00 \times 10^{-12}$ | EFEMP1 |
| 2 | 60175475 | rs1025128 | C | 0.582 | 0.563 | 0.988458 | 1.08 (1.05–1.11) | $2.80 \times 10^{-9}$ | – |
| 2 | 176900271 | rs847139 | C | 0.805 | 0.787 | 0.985197 | 1.11 (1.07–1.14) | $7.20 \times 10^{-11}$ | KIAA1715, EVX2 |
| 2 | 218128152 | rs1863190 | T | 0.78 | 0.76 | 0.997145 | 1.12 (1.08–1.15) | $5.40 \times 10^{-13}$ | DIRC3 |
| 3 | 124450081 | rs4678145 | G | 0.88 | 0.868 | 0.999293 | 1.12 (1.08–1.16) | $4.10 \times 10^{-9}$ | ITGB5, UMPS, KALRN, MUC13 |
| 4 | 13221747 | rs6843953 | T | 0.154 | 0.138 | 0.99611 | 1.14 (1.10–1.18) | $5.80 \times 10^{-12}$ | – |
| 6 | 31440651 | rs3828889 | C | 0.748 | 0.732 | 0.996908 | 1.09 (1.06–1.12) | $1.70 \times 10^{-8}$ | – |
| 6 | 85715955 | rs62422907 | G | 0.899 | 0.887 | 0.991783 | 1.13 (1.09–1.18) | $2.20 \times 10^{-9}$ | – |
| 7 | 44145178 | rs55841377 | C | 0.789 | 0.773 | 0.992012 | 1.09 (1.06–1.13) | $8.40 \times 10^{-9}$ | AEBP1, PLD2, MYL7, GCK |
| 7 | 150542515 | rs6977081 | G | 0.685 | 0.668 | 0.97855 | 1.08 (1.05–1.11) | $1.20 \times 10^{-8}$ | AOC1, TMEME176A, KCNH2, TMEM176B |
| 14 | 76245906 | rs72725608 | C | 0.051 | 0.044 | G | 1.20 (1.13–1.27) | $1.10 \times 10^{-8}$ | IFT43, TTLL5, TGFB3 |
| 15 | 67034812 | rs1866745 | A | 0.369 | 0.35 | 0.993487 | 1.09 (1.06–1.12) | $4.20 \times 10^{-10}$ | SMAD6 |
| 15 | 100692953 | rs72755233 | A | 0.128 | 0.112 | G | 1.18 (1.13–1.22) | $2.30 \times 10^{-15}$ | ADAMTS17 |
| 19 | 8670147 | rs62621197 | T | 0.045 | 0.036 | 0.924188 | 1.31 (1.22–1.40) | $7.50 \times 10^{-14}$ | ADAMTS10, MYO1F |

SNP single-nucleotide polymorphism
[a]Based on NCBI Genome Build 37 (hg19)
[b]The effect allele frequency in cases
[c]The effect allele frequency in controls
[d]The SNP Info score for imputed SNPs; G = genotyped SNP
[e]Genes mapped to these loci based on positional mapping in FUMA (see 'Methods')

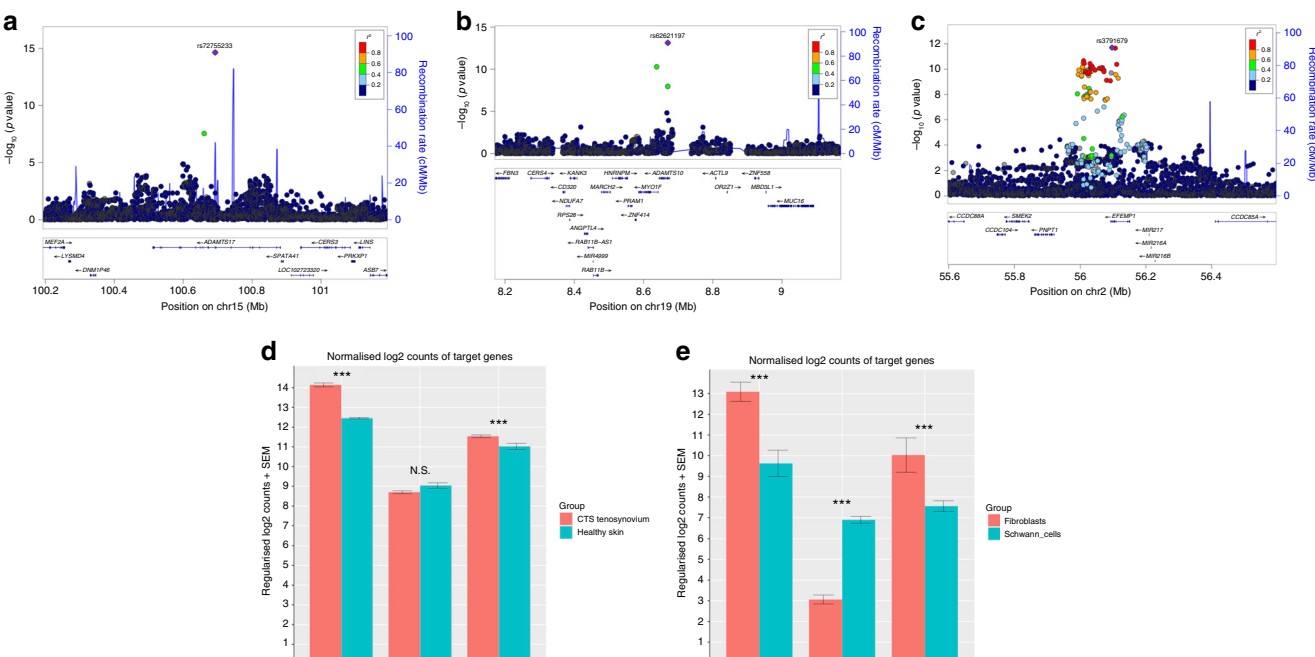

**Fig. 2** Regional association plots for three GWAS loci and the expression of target genes from RNA-Seq. **a** Chromosome 15q26.3 (the site of missense variant in *ADAMTS17*). **b** Chromosome 19p13.2 (the site of missense variant in *ADAMTS10*). **c** Chromosome 2p16.1 (the site of intronic variant in *EFEMP1*). SNP position is shown on the x-axis, and the strength of association on the y-axis. The linkage disequilibrium (LD) relationship between the lead SNP and the surrounding SNPs is indicated by the colour. In the lower panel of each figure, genes within 500 kb of the index SNP are shown. The position on each chromosome is shown in relation to Human Genome build hg19. **d** Comparison of gene expression between tenosynovium of 41 CTS cases and the index finger skin of six healthy individuals. **e** Comparison of gene expression between human cultured fibroblasts and Schwann cells from Weiss et al.[38]. Error bars represent the standard error of the mean of the regularised log2 counts. p Value was determined using Wald test and was FDR adjusted. ***p Value < 0.01; NS = not significant. Source data are provided as a Source Data file. CTS, carpal tunnel syndrome; FDR, false discovery rate; GWAS, genome-wide association study; RNA-Seq, RNA sequencing; SNP, single-nucleotide polymorphism

resides in an enhancer region of *EFEMP1*[27]. A further 3 out of 428 genome-wide significant SNPs had a RegulomeDB score of 2b, which is likely to affect protein binding (Supplementary Data 2).

**Gene mapping**. Functionally annotated SNPs were then mapped to genes based on genomic position and annotations obtained from ANNOVAR, using positional mapping in FUMA. This mapped 25 genes to 12 of the 16 loci (Table 1; Supplementary Fig. 2). A gene-based association analysis was also implemented in MAGMA, and this identified 17 genes that were significantly associated with CTS (Supplementary Table 2; Supplementary Fig. 3), which included 7 of the 25 genes that mapped to our associated loci in FUMA. MAGMA was also used to perform gene-property analysis across 53 GTEx[28] v6 tissue types—the top five expressions were in tibial artery, coronary artery, tibial nerve, aorta and transformed fibroblasts (Supplementary Fig. 4).

Gene-set analysis of the 25 FUMA-mapped genes revealed a strong enrichment for gene ontologies for cellular components associated with the extracellular matrix (adjusted $p = 2.7 \times 10^{-8}$) and GWAS catalogue-reported genes for waist circumference and height (Supplementary Table 3). A similar analysis on the 17 genes prioritised in the MAGMA gene-based analysis also showed strong enrichment for GWAS catalogue-reported genes for height and implicated three further genes in this pathway: *SIN3A*, *PTPN9* and *ZBTB3*. We employed another computational method (XGR[29]) to perform a gene-based enrichment analysis on the 25 FUMA-mapped genes and confirmed this strong enrichment for extracellular matrix-related genes—the greatest enrichment was seen in the "genes encoding core extracellular matrix including ECM glycoproteins, collagens and proteoglycans" ($Z$ score = 3.24, $p = 0.0016$) (Supplementary Table 4).

We performed summary data-based Mendelian randomisation[30] (SMR) analysis using the top associated expression quantitative trait loci (eQTL) in fibroblasts as an instrumental variable to test for the association between the expression level of each gene and CTS. As there are no publicly available eQTL data for tenosynovium (a tissue implicated in CTS pathogenesis), we chose to use eQTL data for transformed fibroblasts from GTEx v7 on the basis that fibroblasts are the principal cellular component of tenosynovium and the significant enrichment for extracellular matrix-related genes in the gene-set analyses. A conservative significance threshold was set at $p < 1.16 \times 10^{-5}$ (0.05/4324 genes). Two genes met the SMR $p$ value significance threshold: *LTBP1* (which mapped to the GWAS locus at 2p22.3) and *MAN2C1* at 15q24.2 (Supplementary Table 5). To exclude SMR associations due to linkage (i.e. two causal variants in LD, with one variant affecting gene expression and the other affecting CTS risk), we performed HEIDI (heterogeneity in dependent instruments) analysis on the two significant genes—both passed the HEIDI test ($p > 0.025$), suggesting an association with CTS through pleiotropy rather than LD and co-localisation.

**SNP-based heritability and partitioned heritability analyses**. Using LDSC regression[24], we calculated the SNP-based heritability estimate of CTS from this GWAS to be 2.4% (SE = 0.17%). We next partitioned the SNP-based heritability across 24 functional genomic categories[31]. There was a significant enrichment of SNP-based heritability in genomic regions conserved across 29 mammal species (10.8-fold enrichment, $p = 0.0013$)—conservation in this context suggests important functional roles and that a significant portion of the common variants associated with CTS are under negative selection pressure[32]. The other statistically significant enrichments implied regulatory activity, including several histone markers (Supplementary Fig. 5).

We also implemented LDSC regression applied specifically to expressed genes (LDSC-SEG)[33] to identify tissues and cell types that are enriched for CTS heritability. Notably, the most significant enrichment was seen in osteoblasts ($p = 5.4 \times 10^{-4}$), followed by smooth muscle myocytes ($p = 2.9 \times 10^{-3}$). By categorising the 205 tested cells and tissues into the nine tissue categories used by Finucane et al.[33], 8/19 of the most significantly enriched tissues and cell types for CTS heritability (all of those that met significance at $p < 0.05$) were within the 'Musculoskeletal/Connective' tissue category (Supplementary Table 6).

**Genetic correlations with CTS-associated phenotypes**. We also used LDSC regression to estimate the degree of correlation between the polygenic architecture of CTS and diseases known to be associated with CTS using publicly available GWAS summary statistics in LD Hub[34], which include the following: rheumatoid arthritis, diabetes, obesity and gout (using serum urate levels as a proxy for the latter). Given that osteoblasts had the top enrichment for partitioned CTS heritability across different cell and tissue types, we also included five bone mineral density traits from LD Hub. Body mass index ($r_{\mathrm{g}} = 0.346$, $p = 5.8 \times 10^{-23}$) and several other measures of obesity were significantly correlated with CTS at a Bonferroni-corrected threshold for significance. Rheumatoid arthritis ($r_{\mathrm{g}} = 0.113$, $p = 9.8 \times 10^{-3}$), serum urate ($r_{\mathrm{g}} = -0.099$, $p = 0.067$) and type 2 diabetes ($r_{\mathrm{g}} = 0.141$, $p = 0.011$) showed weaker correlations, although HbA1c, a measure of long-term plasma glucose concentration, had a significant positive correlation ($r_{\mathrm{g}} = 0.282$, $p = 7.1 \times 10^{-5}$). Lumbar spine bone mineral density (PMID 22504420) had a significant positive correlation with CTS ($r_{\mathrm{g}} = 0.183$, $p = 0.0002$). Notably, measures of height were significantly negatively correlated with CTS ($r_{\mathrm{g}} = -0.217$, $p = 3.7 \times 10^{-9}$ for "Height_2010" and $r_{\mathrm{g}} = -0.223$, $p = 2.1 \times 10^{-6}$ for "Extreme height") (Supplementary Table 7). Consistent with this negative genetic correlation between height and CTS, the four most significantly associated SNPs in this GWAS (rs72755233, rs62621197, rs1863190, and rs3791679) have been previously identified as GWAS loci for height[35-37]. For all but rs1863190, the risk allele for CTS was associated with a lower height in the previous reports. We therefore hypothesised that CTS cases would be shorter than controls within the UK Biobank cohort. We calculated the mean standing height separately for male and female cases and controls, and found that on an average, CTS patients are ~2 cm shorter than controls in both sexes (Table 2).

**Mendelian randomisation study of height and CTS**. We investigated whether there is a causal relationship between height and CTS by performing a two-sample Mendelian randomisation (MR) analysis, using height as the exposure and CTS status as the outcome. We selected SNPs as instrumental variables for height from a large meta-analysis of adult-height GWAS[37]. Using the inverse variance-weighted (IVW) MR method on 601 SNPs, we identified that a 1 SD (equivalent to 9.24 cm) increase in genetically instrumented height was associated with an OR of 0.79 (95%

**Table 2 Comparison of standing height between CTS cases and controls**

| Sex | Case[a] | Control[a] | Difference | $p$ value[b] |
|---|---|---|---|---|
| Male | 173.8 (6.7) | 175.9 (6.7) | 2.1 | $5.53 \times 10^{-80}$ |
| Female | 160.7 (6.2) | 162.7 (6.2) | 2.0 | $1.84 \times 10^{-180}$ |

*CTS* carpal tunnel syndrome
[a]Height is given in cm and the standard deviation is shown in parentheses
[b]Unpaired two-tailed $t$ test

**Table 3 Mendelian randomisation study of height and CTS**

| | Method | Odds ratio (95% CI) of CTS per 1-SD higher height | *p* value |
|---|---|---|---|
| Main analysis | IVW | 0.79 (0.74, 0.83) | $2.24 \times 10^{-15}$ |
| | Weighted median | 0.80 (0.75, 0.86) | $2.33 \times 10^{-9}$ |
| | MR-Egger[a] | 0.72 (0.61, 0.84) | $2.87 \times 10^{-5}$ |
| Sensitivity analysis | IVW | 0.80 (0.75, 0.85) | $9.88 \times 10^{-15}$ |
| | Weighted median | 0.81 (0.76, 0.87) | $7.86 \times 10^{-9}$ |
| | MR-Egger[b] | 0.74 (0.64, 0.86) | $7.89 \times 10^{-5}$ |

The main analysis included 601 SNPs from Wood et al.[37], and the sensitivity analysis was performed after removing 5 SNPs that were significant in both the height and CTS GWAS. The mean height for the whole cohort (both sexes combined) was 168.7 cm; SD = 9.24 cm
*CTS* carpal tunnel syndrome, *GWAS* genome-wide association study, *IVW* inverse variance-weighted
[a]MR-Egger intercept (95% CI): 0.00296 (−0.00163, 0.00756); *p* = 0.21
[b]MR-Egger intercept (95% CI): 0.00244 (−0.00194, 0.00681); *p* = 0.28

CI: 0.74–0.83, $p = 2.24 \times 10^{-15}$) for the development of CTS. The MR-Egger analysis gave an OR of 0.72 (95% CI: 0.61–0.84, $p = 2.87 \times 10^{-5}$) and no evidence to support the presence of confounding by unbalanced genetic pleiotropy (intercept = 0.0030; 95% CI: -0.0016, 0.0076, $p = 0.21$) (Table 3). We performed a sensitivity analysis by removing five SNPs that were genome-wide significant in both the height and the CTS GWAS, and this yielded similar values in the analyses. The funnel plots showed a near-symmetrical distribution of individual variants around the estimate (Fig. 3c, d). The removal of rs724016 (which had a 1/SE value of ~6 and looked asymmetrical on the funnel plot) in a leave-one-out analysis using IVW had no material effect on the MR estimate. Taken together, these results provide strong evidence that height is inversely causal in the pathogenesis of CTS.

**RNA sequencing in carpal tunnel tenosynovium.** On the basis of this enrichment for extracellular matrix-associated genes in the gene-set analyses, we hypothesised that our top candidate genes would be expressed in the connective tissues within the carpal tunnel. RNA sequencing (RNA-Seq) was performed on RNA extracted from surgically resected tenosynovium surrounding the flexor tendons within the carpal tunnel in 41 patients undergoing carpal tunnel decompression surgery. The three genes that were our top candidate genes on the basis of functional annotation (*ADAMTS17*, *ADAMTS10*, and *EFEMP1*) were consistently expressed in CTS tenosynovium, and their expression was above the median (438.6) of library size-normalised read pairs. As we could not surgically resect tenosynovium from healthy controls, we compared the expression of these genes against the skin from the index finger of six healthy (i.e. non-CTS) individuals to ascertain whether gene expression was greater in our tissue of interest. *ADAMTS10* (log2 fold change (lfc) = 0.65) and *EFEMP1* (lfc = 2.29) were significantly upregulated in CTS tenosynovium compared with healthy skin (adjusted *p* value = $2.6 \times 10^{-3}$ and $1.9 \times 10^{-14}$, respectively), although *ADAMTS17* was not (Fig. 2d). Furthermore, using the data from Weiss et al.[38], we found that all the three candidate genes were expressed in both cultured human fibroblasts and Schwann cells, with *ADAMTS17* showing significantly greater expression in Schwann cells (Fig. 2e). We have, therefore, demonstrated the proof of principle that these candidate genes are highly expressed in carpal tunnel tenosynovium, cultured Schwann cells, and fibroblasts, and may therefore have a functional role in these tissues.

**Genetic risk score for CTS.** We calculated the mean weighted genetic risk score (wGRS) for four subgroups of individuals who were included in the GWAS, which are as follows: (1) all CTS cases, (2) all controls, (3) CTS cases with at least one operation code (either OPCS (Office of Population Censuses and Surveys Classification of Interventions and Procedures) or self-reported),

and (4) CTS cases with no operation codes. As expected, the wGRS in CTS cases (1.620) was higher than that in controls (1.566) ($p < 0.0001$). We hypothesised that the CTS cases who have undergone surgery are phenotypically more severe than those CTS cases who have not and would therefore have a higher genetic risk score. Consistent with this, we found a higher wGRS in the operated group compared with the unoperated group (1.622 vs 1.586, $p = 1.5 \times 10^{-4}$) (Table 4). This provides evidence that the wGRS derived from our GWAS hits correlates with disease severity.

**Discussion**

We have identified 16 independent association signals that reached genome-wide significance in our GWAS of 12,312 cases of CTS and 389,344 controls from UK Biobank. We discovered strong evidence of functionality across several of the associated loci and that several histone markers are enriched in CTS-associated regions of the genome. Of the genes implicated in the GWAS, we identified three top candidate genes on the basis of the association *p* value, likely functional consequence of the variant, and biological plausibility, which are *ADAMTS17*, *ADAMTS10* and *EFEMP1*. We found enrichment for musculoskeletal and connective tissues when we partitioned the heritability of CTS across different cell and tissue types, and this was supported by the genetic correlation between CTS and anthropometric phenotypes (body mass index and height). We established the causality between shorter height and higher risk of CTS by Mendelian randomisation and, consistent with this, we found that patients with CTS are, on an average, ~2 cm shorter than controls. Gene-based enrichment analyses of the mapped genes demonstrated an enrichment for extracellular matrix components, and we demonstrated high levels of gene expression for the top three candidate genes in tenosynovium resected from CTS patients. We calculated the polygenic susceptibility to CTS using a wGRS based on the associated loci and found a significantly higher risk score in operated CTS cases than in unoperated CTS cases, consistent with the notion that carrying a greater number of the risk alleles leads to a more severe phenotype.

Our rationale for undertaking this study was to better understand the pathogenesis of CTS, and we formulated two principal a priori hypotheses (that are not mutually exclusive), which are as follows: genetic risk for CTS may relate to (1) the carpal tunnel environment through which the median nerve transits or (2) the vulnerability of these nerve fibres to compression. We used a GWAS approach to interrogate the common variants associated with CTS, and our results suggest that genetic factors relating to growth and the extracellular matrix (with likely implications for nerve structure or environment) are the predominant genetic determinants of CTS. These two hypotheses are of course not

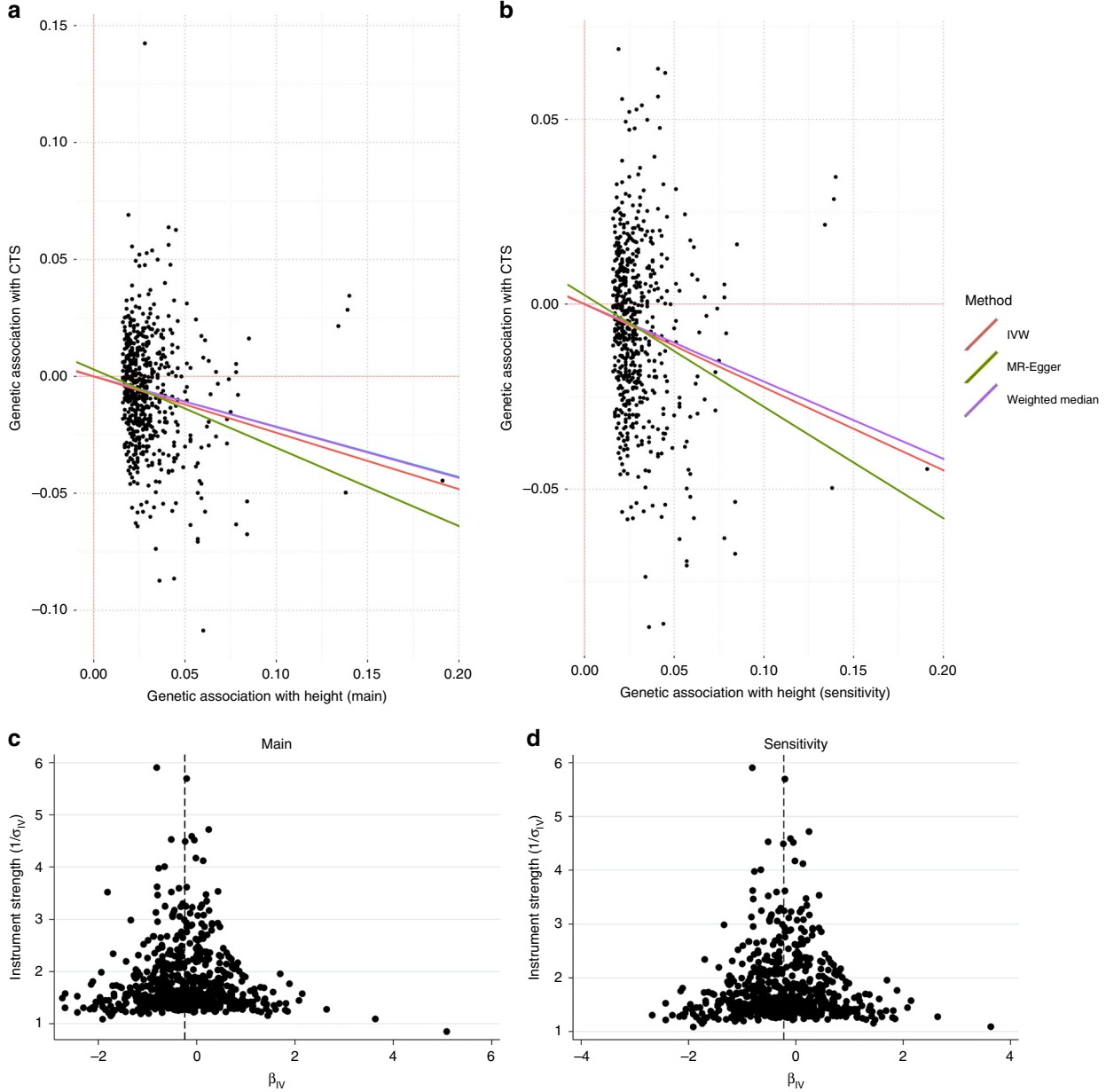

**Fig. 3** Mendelian randomisation (MR) analysis of height and CTS. **a** MR plots for the main analysis (601 SNPs) using IVW, MR-Egger and weighted median methods. **b** MR plots for the sensitivity analysis (596 SNPs), with 5 SNPs that were significantly associated with both height and CTS removed. **c** Funnel plot of instrument strength vs effect size for the main analysis and **d** the sensitivity analysis. The removal of rs724016 (which is the only SNP with a 1/SE value of ~6 in plots (**c**, **d**)) had no material effect on the MR estimates using IVW. Odds ratio for CTS per 1-SD higher height in the main analysis was 0.79 (95% CI 0.75–0.85) and in sensitivity analysis was 0.81 (95% CI 0.76–0.85). CTS, carpal tunnel syndrome; IVW, inverse variance-weighted; SNP, single-nucleotide polymorphism

### Table 4 Weighted genetic risk score in the UK Biobank cohort

| Group | CTS cases | Controls | p Value[b] | CTS cases with operation code | CTS cases without operation code | p Value[c] |
|---|---|---|---|---|---|---|
| N | 12,312 | 389,344 | | 11,626 | 686 | |
| wGRS[a] | 1.620 (0.235) | 1.566 (0.232) | $4.87 \times 10^{-136}$ | 1.622 (0.235) | 1.586 (0.235) | $1.45 \times 10^{-4}$ |

CTS carpal tunnel syndrome, wGRS weighted genetic risk score
[a]Standard deviation shown in parentheses
[b]Unpaired two-tailed t test between CTS cases and controls
[c]Unpaired two-tailed t test between CTS cases with operation code and CTS cases without operation code

mutually exclusive; for example, *COL11A1* and *EFEMP1* are highly expressed in both transformed fibroblasts and in tibial nerve within the GTEx dataset (Supplementary Table 8). This suggests that alterations within the connective tissue environment of the nerve itself may contribute to the predisposition to CTS.

Our top two significantly associated SNPs in the GWAS are missense variants in *ADAMTS17* and *ADAMTS10*. The ADAMTS proteins are a family of multidomain, secreted extracellular proteolytic enzymes related to matrix metalloproteinases (MMPs), which are involved in extracellular matrix maintenance via the cleavage of procollagen and proteoglycans. Thus, they participate in various cellular processes, including cell adhesion and migration[39]. Although the substrates of ADAMTS10 and ADAMTS17 are yet to be characterised[40,41], rs72755233 results in a non-conservative threonine-to-isoleucine substitution in the protease domain of ADAMTS17[35] and is predicted by SIFT to be deleterious. rs62621197 results in an arginine-to-glutamine substitution directly adjacent to the furin cleavage site of ADAMTS10, where it has been suggested that the presence of glutamine may decrease the enzyme's activation efficiency[35,42]. It is quite possible, therefore, that these two SNPs are the causative variants that confer an increased risk to an individual developing CTS.

An association between CTS and anthropometric measurements has been observed previously, including short stature, short hand length, increased palm width, and a greater wrist index (wrist depth/width)[43,44]. It is postulated that having shorter hands and a 'squarer' wrist produces an anatomical configuration that predisposes the median nerve to greater compression[45]. Interestingly, mutations in *ADAMTS10* can manifest as autosomal recessive Weill–Marchesani syndrome (WMS), a rare condition characterised by short stature, brachydactyly, joint stiffness and eye anomalies[46], while mutations in *ADAMTS17* can cause a WMS-like syndrome, in which the affected patients have a short stature and eye anomalies[47]. CTS in association with WMS has been reported previously[48], including cases of children with bilateral CTS[49]. This supports the hypothesis that abnormalities of musculoskeletal growth and development due to the altered activity of these two *ADAMTS* genes might contribute to CTS aetiology. Furthermore, we have provided, for the first time, an explanation for the observation that CTS patients are shorter in stature than healthy controls.

In order to further interrogate the relationship between height and CTS, we performed an MR analysis. The results of this analysis provide strong evidence of the inverse causal effect of height on CTS status, and this relationship persisted in various sensitivity analyses. This analysis, therefore, supports the hypothesis that altered anthropometric measures (of which height is an easily measured proxy) contribute to CTS predisposition. This is further supported by the genetic correlation between CTS and other anthropometric variables, and in the partitioned analysis of CTS heritability across different cell and tissue types, in which the top enrichment was found in osteoblasts, and nearly half of the top enriched cell and tissue types were in the 'Musculoskeletal/Connective tissue' category.

Apart from having an effect on musculoskeletal development, variants in extracellular matrix components may also play a role within the median nerve. Consistent with this, gene-property analysis in MAGMA found that tibial nerve and transformed fibroblasts were the two tissues that showed high enrichment (ranked third and fifth, respectively, out of the 53 tissue types). Schwann cells form a basement membrane composed of extracellular matrix, which is required for myelination[50]. A previous study of an experimental nerve injuries in a mouse model showed altered expression of a number of genes following injury[51]. Interestingly, *ADAMTS17* is upregulated in the distal nerve stump following nerve crush, and this expression is dependent on the expression of neuregulin-1, a growth factor that has

a critical role in remyelination and repair. Moreover, *ADAMTS17* is more highly expressed in human cultured Schwann cells than in fibroblasts (Fig. 2e). This suggests the intriguing hypothesis that variation in *ADAMTS17* may impede the neural recovery following carpal tunnel treatment and deserves further study.

The fourth most significantly associated SNP in the GWAS, rs3791679, resides in an enhancer region of *EFEMP1*[27] (EGF-containing fibulin-like extracellular matrix protein 1) and has a CADD score of 14.94, suggesting that it is a plausible causal variant at this locus. It is an eQTL for *EFEMP1* in human skin and thyroid tissue, and several other SNPs at this locus in LD with this SNP are also eQTLs for *EFEMP1* (Supplementary Table 9). *EFEMP1* is a gene that encodes protein fibulin-3, an extracellular matrix glycoprotein associated with basement membranes, elastic fibres and matrix components[52]. *EFEMP1* is implicated in extracellular matrix architecture[36] as well as in bone- and cartilage-development pathways (hence its potential role in contributing to the observed enrichments between CTS and height, osteoblasts and lumbar spine bone mineral density). Variants at the *EFEMP1* locus have been associated with adult height[36], inguinal hernia[53], and various cancers[54]; *EFEMP1* knockout mice demonstrated disruption or reduction in elastic fibres in their connective tissues[55].

Similarly, our top GWAS hit at 1p21.1 is a variant approximately 0.1 Mb downstream of *COL11A1* (rs12406439, $p = 1.1 \times 10^{-8}$, OR = 1.05), the only protein-coding gene within 0.5 Mb of this SNP (Supplementary Fig. 7). *COL11A1* encodes α1(XI), one of the three chains that constitute the triple helix in type XI collagen[56]. Type XI collagen has been implicated in a range of musculoskeletal conditions, including lumbar disc herniation[57], Achilles tendinopathy[58], and CTS[59]. In the latter study, homozygotes of the T allele at rs3753841 (a missense mutation) in *COL11A1* were found to be significantly over-represented in CTS patients vs controls. rs3753841 is approximately 140 kb from our top SNP at this locus (rs12406439), and the two SNPs are in moderate linkage disequilibrium ($r^2 = 0.69$; 1000 Genomes Project Phase 3, GBR (British in England and Scotland) data). In fact, rs3753841 is a genotyped SNP in the UK Biobank array, and its *p* value in our association study was $5.4 \times 10^{-5}$, which is suggestive of association with CTS. Taken together, these data strongly suggest that *COL11A1*, like *EFEMP1*, is another candidate connective tissue gene, the abnormal expression or function of which contributes to the aetiology of CTS.

The subsynovial connective tissues (SSCTs) within the carpal tunnel have previously been proposed to play an important role in the pathophysiology of CTS[59,60], and the SSCTs from CTS patients show evidence of fibrosis[12,13]. The fibrosis and thickening of SSCT may result in an increased pressure acting on the median nerve, which may in turn interfere with neural biomechanics[61] and microcirculation[62], which initiates downstream mechanisms such as demyelination[63], impaired axonal transport[64] and altered nerve conduction[65]. Thus, aberrations in elastic and connective tissue components in the extracellular matrix are likely to contribute to this pathophysiological process, and this theory was supported by our gene-based enrichment analysis. We hypothesise that the risk allele at rs3791679 might lead to the overexpression of *EFEMP1* in synovium and may contribute to increased tethering and impaired gliding of the median nerve, thus predisposing it to shear stress and subsequent fibrosis. Interestingly, miR-338-5p has recently been shown to inhibit the synthesis of *EFEMP1* by directly binding to the *EFEMP1* mRNA 3′-UTR region in glioblastoma-derived cells[66], illustrating the potential of this pathway as a modifiable therapeutic target.

Transforming growth factor-beta (TGF-β) plays a critical role in the regulation of extracellular matrix gene expression[67], and several previous studies have reported the role played by TGF-β and

the TGF-β/Smad signalling pathway in the development of SSCT fibrosis in CTS[68,69], including the finding that the inhibition of TGF-β1 in SSCT fibroblasts from CTS patients resulted in the down-regulation of several fibrosis-related genes and the inhibition of Smad activity[68]. It is striking that three of the genes that mapped to our associated loci were part of the TGF-β/Smad signalling pathway: latent transforming growth factor beta-binding protein 1 (*LTBP1*), transforming growth factor β-3 (*TGFB3*) and *SMAD6*. LTBP1 regulates TGF-β through its interaction with the extracellular matrix, and, notably, *LTBP1* was one of only two genes that met Bonferroni-corrected significance in the SMR analysis using eQTL data from transformed fibroblasts. Local corticosteroid injection is an effective therapy for CTS[70], and dexamethasone has been shown to upregulate TGF-β3 (but not TGF-β1 or TGF-β2) receptors in hepatic stellate cells and myofibroblasts[71]. It will therefore be interesting to determine in future studies whether corticosteroids can also regulate the expression of the other extracellular matrix genes implicated in this GWAS.

It is noteworthy that our GWAS did not appear to implicate any gene that is known to confer risk to CTS in the context of inherited neuropathies. Hereditary neuropathy with liability to pressure palsy (HNPP) is a monogenic disorder that illustrates the potential relevance of nerve vulnerability in the predisposition to CTS. HNPP is caused by the haplo-insufficiency of *PMP22* gene[72]; reduced expression of this myelin protein results in myelin instability and conduction block in response to external pressures[73], and CTS is a common manifestation in patients with HNPP[74]. Similarly, mutations in *SH3TC2* are associated with Charcot–Marie–Tooth disease, which also confers susceptibility to neuropathies, including CTS[75]. Therefore, these two genes would have been reasonable a priori candidate genes for CTS susceptibility in the general population, but neither *PMP22* nor *SH3TC2* was enriched within our GWAS. In contrast, there was a marked over-representation of growth- and extracellular matrix-related genes, and this may in part explain why the only statistically significant genetic correlations with CTS that withstood corrections for multiple testing were anthropometric phenotypes and not rheumatoid arthritis, principally an autoimmune disease, or diabetes, principally a metabolic disease.

One clear limitation of this study is the absence of a replication cohort, meaning that systematic biases within our cohort may lead to the observed associations. However, several factors mitigate this deficiency. Firstly, using a large resource such as UK Biobank has allowed us to identify a substantial cohort of over 12,000 CTS cases and nearly 400,000 controls, a number in considerable excess of most published two-stage GWAS. The statistical power conferred by using a dataset of this size has allowed us to discover multiple genome-wide significant SNPs at each associated locus, even after the application of stringent QC criteria on both genotyped and imputed SNPs. Secondly, our wGRS was associated with disease severity, providing some degree of internal validation. Thirdly, we have previously performed a validation study of the codes used to designate a participant as a 'CTS case' by interrogating the hospital case notes of a subset of UK Biobank participants with CTS[76]; the overall positive predictive value for true clinical disease cases was 94%, lending considerable support to the accuracy of our phenotyping methodology, which is often lacking in other studies. However, the possibility of false-positive association signals from a single-stage GWAS is acknowledged, and replication in a well-powered independent cohort is needed to confirm the associations at our loci. Our attempt to perform a replication in an under-powered independent cohort with a less stringent case definition is documented in the Supplementary Information; in spite of the various limitations, the *ADAMTS10* locus replicated at a Bonferroni-corrected significance threshold and 13/16 loci showed directional concordance between the two GWAS, with an LDSC

regression-computed genetic correlation of 0.90. The relatively modest size of the SNP-based heritability estimate (2.4%) from this single GWAS is in keeping with other highly polygenic traits, and larger sample sizes are needed to discover further genetic loci. Moreover, extending this investigation to populations of non-European ancestry will likely uncover further association signals and yield further insights into the genetic pathophysiology of CTS.

Another limitation is the absence of an ideal control for our gene-expression studies in carpal tunnel tenosynovium, the harvesting of this tissue from healthy controls being unethical. Median nerve-innervated index finger skin from healthy controls has therefore served as a surrogate 'control' tissue, and, as such, the comparison of gene expression between tenosynovium and skin must be interpreted with caution. Despite this limitation, we demonstrated that our top three candidate genes are highly expressed in tenosynovium. Future functional assays will be required to confirm whether the candidate genes identified herein are causally implicated in CTS susceptibility.

In summary, this study revealed 16 genetic loci that underlie CTS, the most common and economically important entrapment neuropathy. Our findings strongly suggest that the genetic susceptibility to CTS arises from altered musculoskeletal growth and development and/or aberrant connective tissue architecture, and implicates an inverse causal role of height in the aetiology of CTS. The insights into the aetiology of CTS have revealed testable functional hypotheses and potential therapeutic avenues for further research.

## Methods

**Ethics approval**. UK Biobank has approval from the North West Multi-Centre Research Ethics Committee (11/NW/0382), and this study (The Genetics of Carpal Tunnel Syndrome) has UK Biobank study ID 10948. The Pain in Neuropathy Study (PiNS) has ethical approval from London Riverside Research Ethics Committee (10/H0706/35).

**Study population and phenotype definition**. For the GWAS, we used the UK Biobank resource, a prospective cohort study of ~500,000 individuals from the UK, aged between 40 and 69 years, who have had whole-genome genotyping undertaken and have allowed the linkage of these data with their medical records[22].

For the association study, CTS cases were identified using diagnosis and operation codes from the UK Biobank showcase, by selecting individuals who had one or more of the following four codes (the codes are shown in parentheses):

1.  ICD-10 code for CTS (G560)
2.  OPCS code for carpal tunnel release (A651) or revision of carpal tunnel release (A652)
3.  Self-reported operation code for carpal tunnel surgery (1501)
4.  Self-reported non-cancer illness code for CTS (1541)

A total of 15,241 participants within the UK Biobank cohort had at least one of the above codes and were classed as cases (Supplementary Table 1). Of these, 12,312 participants passed QC and were included in the GWAS.

For the height analysis, we used UK Biobank Data Field 50 to determine the participants' standing height, as measured at their initial assessment visit. Height data were available for 400,806 individuals who were included in the GWAS.

**Genotyping**. UK Biobank's genotyping, QC and imputation methodology are described in detail elsewhere[22]. Briefly, UK Biobank contains genotypes of 488,377 participants who were genotyped on two very similar genotyping arrays. A total of 49,950 participants were genotyped on the UK BiLEVE Axiom Array (807,411 markers), and 438,427 participants were genotyped using the UK Biobank Axiom Array (825,927 markers); the two arrays share approximately 95% of their marker content. Genotypes were called from the array intensity data, in 106 batches of approximately 4700 samples each using a custom genotype-calling pipeline.

**Quality control**. QC was performed using PLINK v1.9 and R v3.3.1. We initially removed all SNPs with a call rate <90%, accounting for the two different genotyping platforms used to genotype the individuals. We then performed sample-level QC by excluding individuals with one or more of the following: (1) heterozygosity >3 SD from the mean (calculated using UK Biobank's principal component analysis (PCA)-adjusted heterozygosity values, Data Field 20004), (2) discrepancy between genetically inferred sex (Data Field 22001) and self-reported sex (Data Field 31) or individuals with sex chromosome aneuploidy (Data Field 22019), and (3) a call rate <98%. We then excluded individuals who were not in the subset of individuals selected by UK Biobank as having a white British ancestry (on the basis of PCA and

self-reporting as British; Data Field 22006). We merged our data with the publicly available data from the 1000 Genomes Project and performed PCA using flashpca and confirmed that the white British-ancestry individuals from UK Biobank overlapped with the GBR individuals from the 1000 Genomes Project (Supplementary Fig. 1b). As we were using BOLT-LMM in our analysis, we did not exclude any individuals based on relatedness[77]. In total, 86,693 individuals were excluded based on the above criteria. We then performed SNP-level QC, by excluding SNPs with <98% call rate, Hardy–Weinberg equilibrium $p < 10^{-4}$, and an MAF < 1%. A total of 230,562 SNPs were excluded. Finally, we excluded six individuals who harboured an abnormal number of SNPs with a minor allele count of 1, which were visual outliers when autosomal heterozygosity was plotted against call rate. This generated a final set of 401,667 individuals and 547,011 SNPs. A flow chart summarising our QC protocol is shown in Supplementary Fig. 1a.

**Imputation.** A detailed description of the phasing and imputation of SNPs undertaken by UK Biobank is given elsewhere[22]. Briefly, phasing on the autosomes was performed using SHAPEIT3, using the 1000 Genomes Phase 3 dataset as the reference panel. For imputation, both the Haplotype Reference Consortium reference panel and a merged UK10K/1000 Genomes Phase 3 panel were used, resulting in a dataset with 92,693,895 autosomal SNPs, short indels and large structural variants in 487,442 individuals. Imputation files were released in the BGEN (v1.2) file format.

**Association analysis.** Genome-wide association testing was undertaken across 547,011 genotyped SNPs and ~8.4 million imputed SNPs with an MAF ≥ 0.01 and Info score ≥ 0.9, using a linear mixed non-infinitesimal model implemented in BOLT-LMM v2.3[23]. A reference genetic map file for hg19 and a reference LD score file for European-ancestry individuals included in the BOLT-LMM package were used in the analysis. The following covariates were used in the association: genotyping platform and genetic sex. The $\lambda_{GC}$ demonstrated some inflation (1.15), but the LDSC regression intercept[24] of 1.015, with an attenuation ratio of 0.073, indicated that the inflation was largely due to polygenicity and the large sample size. The quantile–quantile plot is shown in Supplementary Fig. 1c. Conditional analysis was performed at each associated locus, using the same covariates as before, plus the allelic dosage of the most significantly associated SNP, computed using QCTOOL v2. Regional LocusZoom[78] plots for our top three candidate SNPs are shown in Fig. 1; plots for the remaining 13 SNPs are shown in Supplementary Fig. 7.

**Replication.** Attempts to replicate the associated SNPs in an independent cohort from the Estonian Genome Centre at the University of Tartu (EGCUT) are described in the Supplementary Discussion.

**Functional annotation of SNPs.** Genomic risk loci were defined from the SNP-based association results using FUMA[25] v1.3.3. Independent significant SNPs were selected on the basis of genome-wide significance ($p < 5 \times 10^{-8}$) and whether independent from each other ($r^2 < 0.3$) within in a 1 Mb window. The UKB Release 2 White British population was selected as the reference population panel, which corresponds to the UK Biobank population used in this GWAS. Functional annotation was carried out in FUMA, with annotations including the ANNOVAR categories, CADD scores, RegulomeDB scores, and chromatin states (Supplementary Data 2). Functionally annotated SNPs were mapped to genes based on the physical position in the genome (FUMA positional mapping), resulting in 25 mapped genes at 12 of the 16 associated loci. Given the high LD and the high gene density in the extended major histocompatibility locus on chromosome 6 (25–33 Mb), this region was excluded from the FUMA annotations.

**Gene-based analysis and gene-property analysis.** A gene-based analysis was performed using MAGMA[79] v1.06, implemented in FUMA—SNPs that were located within 18,633 protein-coding genes were used to derive a $p$ value for the association with CTS status. A Bonferroni correction was applied to control for multiple testing, with a genome-wide significance threshold of $2.68 \times 10^{-6}$. Seventeen genes met this threshold (Supplementary Table 2). MAGMA was also used to perform a gene-property analysis in order to identify particular tissue types relevant to CTS. This analysis determines if tissue-specific differential expression levels are predictive of the association of a gene with CTS, across 53 different tissues taken from the GTEx v6 database (Supplementary Fig. 4).

**Gene-set analyses.** In order to gain insight into the biological systems implicated by our prioritised genes, we implemented gene-set analyses using the GENE2-FUNC tool in FUMA. Two analyses were performed using (1) the 25 genes prioritised by FUMA positional mapping (Table 1) and (2) the 17 genes that met genome-wide significance in the MAGMA gene-based analysis (Supplementary Table 2). The following settings were applied: Benjamini–Hochberg false discovery rate (FDR) for multiple testing correction, adjusted $p$ value cut-off = 0.05, the minimum number of overlapped genes = 5, GTEx v7 RNA-Seq expression data. The genes were tested for over-representation in different gene sets, including Gene Ontology cellular components (MSigDB c5) and GWAS catalogue-reported genes (Supplementary Table 3). We also implemented a similar analysis using XGR

software[29] (gene-based enrichment analysis, selecting the 'canonical pathways' ontology) to identify enriched ontology terms within the 25 genes mapped by FUMA (Supplementary Table 4).

**Summary-based Mendelian randomisation.** We performed SMR and HEIDI analyses[30] to identify genes with expression levels associated with CTS due to pleiotropy, using the summary statistics from the GWAS and eQTL data for transformed fibroblasts from GTEx v7 (Supplementary Table 5). We tested for associations between CTS status and the expression level of each gene using the top associated eQTL of each gene as an instrumental variable. The Bonferroni-corrected significance level for a gene was $P_{SMR} < 2.68 \times 10^{-6}$, i.e. 0.05/4323 genes. We also sought to exclude SMR associations due to linkage by implementing HEIDI. The HEIDI analysis tests whether there is heterogeneity in SMR estimates at SNPs in LD with the top associated eQTL and distinguishes pleitropy from linkage. Genes not rejected by HEIDI ($p_{HEIDI} > 0.05$/number of $p_{SMR}$-significant probes) have evidence of association with the disease through pleiotropy (rather than linkage) at a shared genetic variant.

**SNP-based heritability analyses.** The LD intercept and the mean $\chi^2$ test statistic for the CTS GWAS were calculated using LDSC regression[24], and the attenuation ratio was calculated using the following formula: (LD intercept - 1)/(mean $\chi^2$ - 1). LDSC was also used to estimate the SNP-based heritability for CTS based on our GWAS summary statistics. This method derives the heritability of a phenotype by regressing a SNP's association statistic onto its LD score (the sum of squared correlations between the minor allele count of a SNP and the minor allele count of every other SNP).

Partitioned heritability analysis was performed using stratified LDSC regression with the aim of determining if SNPs that explain the heritability of CTS cluster in functional regions across the genome. We tested for CTS heritability across the 24 functional categories in the 'full baseline model' derived from publicly available annotations that are not specific to any cell type[31]. The enrichment metric is derived by dividing the proportion of heritability captured by the functional annotation by the proportion of SNPs contained within it, and thus it describes whether a particular annotation contains a greater or lesser proportion of the heritability than would be expected based on the proportion of SNPs it contains. We calculated stratified LD scores using European-ancestry samples in the 1000 Genomes Project and only included HapMap3 SNPs with MAF > 0.05.

We further performed LDSC regression applied specifically to expressed genes (LDSC-SEG) to identify CTS-relevant tissues and cell types[33]. We analysed 205 different tissues or cell types from publicly available GTEx[28] RNA-Seq and Franke Lab human, mouse and rat array data[80,81]. We used the classification of the 205 tissue and cell types into nine distinct categories of related phenotypes, as illustrated by Finucane et al.[33]. A positive regression coefficient for the annotation signifies a positive contribution of this annotation to trait heritability; this regression coefficient is given (along with its standard error and $p$ value) in Supplementary Table 6.

**LD score regression and genetic correlations.** Genetic correlation ($r_g$) values between CTS and phenotypes known to be associated with CTS (rheumatoid arthritis, diabetes, obesity and elevated urate, as a proxy measure for gout) were also computed using LDSC[24,82], implemented on LD Hub[34]. We harmonised our CTS GWAS summary statistics with the LD-pruned list of well-imputed SNPs (with SNPs in the major histocompatibility region removed) on the LD Hub web interface, leaving a total of 1,171,431 SNPs.

For the genetic correlation study, we used publicly available GWAS meta-analysis results available on the LD Hub interface and selected the following trait groups for analysis: "Autoimmune diseases", "Anthropometric traits", "Uric acid", "Glycemic traits", and "Bone mineral density" (the latter was selected on the basis that skeletal growth and osteoblasts were implicated in CTS pathogenesis in the preceding analyses). We removed all but "Rheumatoid Arthritis" from the "Autoimmune diseases" trait group and also removed "Body fat" from the "Anthropometric traits" group, as this meta-analysis included participants of non-European ancestry, leaving a total of 36 traits (Supplementary Table 7). We performed a Bonferroni correction for multiple testing by applying a $p$ value threshold of 0.05/36 = 0.0014 to define a correlation as statistically significant.

**Mendelian randomisation.** We used MR to assess the potential causal role of height in CTS. A two-sample MR approach was taken using genetic effect estimates for SNPs that are genome-wide significant for height, and the corresponding effect estimates with risk of CTS. SNPs associated with height were taken from the 2014 GIANT consortium meta-analysis of 253,288 individuals of European ancestry[37], which identified 697 variants associated with adult height. We purposefully did not use a more recent meta-analysis of adult height by the GIANT consortium[83], as this study includes ~500,000 UK Biobank participants and would introduce bias from significant sample overlap with the CTS GWAS. We selected independent SNPs ($r^2 < 0.1$) from the raw meta-analysis data that were associated with height at GWAS significance ($p < 5 \times 10^{-8}$) and were also present in our CTS GWAS summary statistics. A total of 601 independent SNPs were selected on this basis, and we manually checked the strand alignment between the two sets of summary statistics and the agreement of effect allele frequencies.

We performed the analysis using the MendelianRandomization package for R[84]. This calculated Wald ratios ($\beta_{IV}$) and corresponding 95% CIs for each SNP. The ratios were calculated by dividing the per-allele log-OR of CTS ($\beta_{ZY}$) by the per-allele difference in height for each SNP ($\beta_{ZX}$). We performed three different MR analyses: (1) IVW MR method, which assumes all SNPs are valid instrumental variables. As such, conventional linear regression analysis of the Wald ratios for each SNP was undertaken and weighted by the inverse variance of the IV estimate. This method constrains the regression when $\beta_{ZX}$ is equal to zero and $\beta_{ZY}$ is also zero. (2) MR-Egger regression to test for horizontal pleiotropy and give a causal estimate in the presence of such pleiotropy. As in the IVW method, $\beta_{ZY}$ is again plotted against $\beta_{ZX}$. However, the intercept is not fixed and, therefore, the deviation from the origin gives evidence for pleiotropic effects in the corresponding direction. In the absence of unbalanced pleiotropy, if SNPs are viewed individually, the $\beta_{IV}$ values will be symmetrically distributed around the point estimate, as demonstrated by a funnel plot. Besides indicating pleiotropy, the MR-Egger provides a causal estimate for the effect of height on CTS when the exclusion restriction assumption is relaxed. (3) The weighted median approach was also used. In this, MR estimates are ordered and weighted by the inverse of their variance. Provided more than 50% of the total weight comes from SNPs without pleiotropic effects, the median MR estimate should remain unbiased to pleiotropy. Hence, the weighted median approach is more robust to violation of the InSIDE assumption than MR-Egger[85].

We also performed a sensitivity analysis by repeating the above three analyses by excluding 5/601 SNPs that were genome-wide significant in both the height and CTS GWAS (or was a proxy SNP in high LD), which are as follows: rs7517682, rs6751657, rs3791679, rs17181956, and rs6955948. A SNP that appeared asymmetrical on the funnel plot (rs724016) was removed as a further sensitivity analysis in a leave-one-out analysis, and this had no material effect on the MR estimate (Fig. 3c, d).

**Genetic risk score.** wGRS was calculated using the method described by de Jager et al.[86]. The number of risk alleles per SNP (i.e. SNP dosage) was multiplied by the weight for that SNP, according to the following formula:

$$wGRS = \sum_{i=1}^{n} W_i X_i$$

where $i$ is the SNP, $n$ is the number of SNPs (16 in this case), $W_i$ is the weight for the SNP (the natural logarithm of the odds ratio for that allele), and $X_i$ is the number of effect alleles. The number of effect alleles was calculated as a SNP dosage, using QCTOOL v2. Computation of wGRS and statistical testing between the different subgroups was carried out in R. To compare operated vs unoperated CTS cases, we selected CTS cases within our post-QC UK Biobank cohort who had at least one operation code ($n = 11,626$) and compared their wGRS with those of the CTS cases without any operation code ($n = 686$).

**Biological samples.** For the gene-expression studies, we used tissues from the participants recruited to PiNS. All participants with CTS recruited to PiNS have clinically and electrophysiologically diagnosed CTS. Of the 41 participants who donated tenosynovium for this study, there were 14 males and 27 females, with a mean age of 63.8 years (SD 11.9) and a median electrodiagnostic severity score of 3 (moderate CTS), interquartile range = 1. Index finger skin was obtained from six healthy participants (two males and four females; mean age 64.5 years, SD 9.47), without evidence of CTS (an electrodiagnostic severity score of 0).

We surgically resected carpal tunnel tenosynovium during carpal tunnel decompression surgery. Skin samples from six control participants were collected on the volar aspect of the index finger using a 3 mm punch biopsy. Synovial specimens were immersed in RNAlater™ Stabilization Solution (Thermo Fisher Scientific), whereas the skin biopsies were snap frozen in liquid nitrogen; both were stored at -80 °C. RNA was extracted using a combination of phenol extraction and column purification[87]. Briefly, samples were homogenised in Trizol (Invitrogen), mixed with chloroform and centrifuged, and the aqueous liquid phase was collected. This solution was placed into High Pure RNA Isolation Kit columns (Roche) and RNA purified and eluted according to the manufacturer's instructions.

**RNA-Seq and analysis.** Library preparation was polyA enriched and directional, using Illumina TruSeq Stranded mRNA and standard universal Illumina multiplexing adaptors: 5′-P-GATCGGAAGAGCACACGTCT and 5′-ACACTCTTTCCCTACACGACGCTCTTCCGATCT. The polyA-selected RNA was converted to cDNA using the strand-specific dUTP strand-marking protocol[88]. Amplification was performed by unique dual indexing using indexing primers described elsewhere[89]. Sequencing was performed using HiSeq4000 with 75 bp read length and paired-end reads. Healthy skin samples and CTS tenosynovium samples were extracted and sequenced using the same protocol but in different library batches.

FastQ sequencing files were produced that encode quality metrics following the Sanger standard, i.e. Sanger qualities, using the standard Phred score[90]. All sequencing lanes gave a high-yield and consistent GC content, consistent and expected sequence inserts between the paired-end adaptors and high-quality base calling. Individuals were multiplexed in lanes.

Mapping to the genome was done using STAR aligner[91]. Reads were mapped on the GRCh38 human genome. STAR was run using default parameters. The genome was indexed using −sjbdOverhang (Read mate length 1). Counts were assigned using HTSeq[92]. Differential expression analysis was performed using DESeq2[93]. The significance cut-off was FDR-adjusted $p$ value < 0.05. Supplementary Fig. 6 shows the hierarchical clustering of the 41 CTS tenosynovium samples and six index finger skin samples.

**Analysis of fibroblast vs Schwann-cell RNA-Seq data.** RNA-Seq data for cultured human fibroblasts ($n = 3$) and Schwann cells ($n = 5$) published by Weiss et al.[38] were downloaded from GEO (GSE90711). Reads were mapped to GRCh38.88 human genome using STAR aligner[91] with default settings. Counts were assigned to GRCh38.88 ENSEMBL gene annotation using HTSeq[92]. Differential expression analysis was carried out using DESeq2[93] with a zero-mean normal prior and log2 fold-change shrinkage. For visualisation purposes, counts were normalised and transformed using the regularised log2 transformation plus/minus the standard error of the mean. Differential gene expression for our top three candidate genes (ADAMTS17, ADAMTS10, EFEMP1) is shown in Fig. 2 and in Supplementary Table 8.

**URLs.** URLs for online resources referenced in the manuscript can be found at UK Biobank, www.ukbiobank.ac.uk/; BOLT-LMM, https://data.broadinstitute.org/alkesgroup/BOLT-LMM/; CADD, https://cadd.gs.washington.edu/; RegulomeDB, http://www.regulomedb.org/; ANNOVAR, http://annovar.openbioinformatics.org/en/latest/; FUMA, http://fuma.ctglab.nl/; MAGMA, https://ctg.cncr.nl/software/magma; GTEx Portal, http://gtexportal.org/home/; XGR, http://galahad.well.ox.ac.uk:3020/; LD Hub, http://ldsc.broadinstitute.org/ldhub/; SIFT, https://sift.bii.a-star.edu.sg/; PLINK, http://pngu.mgh.harvard.edu/~purcell/plink/; R, https://www.r-project.org; 1000 Genomes Project, http://www.1000genomes.org; flashpca, https://github.com/gabraham/flashpca/; SHAPEIT3, https://jmarchini.org/shapeit3/; HRC, http://www.haplotype-reference-consortium.org/; and QCTOOL v2, http://www.well.ox.ac.uk/~gav/qctool_v2/#overview.

## Data availability

Full UK Biobank data are available by the direct application to UK Biobank (see URLs). The RNA-Seq data have been uploaded to GEO (accession no. GSE "108023"). Full GWAS summary statistics can be found in Supplementary Data 3. All the other relevant data are available from the authors on request.

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

## Acknowledgements

We are grateful to all the individuals who participated in the study. This research has been completed using the UK Biobank Resource under the application 10948. D.F. was supported by an Intermediate Clinical Fellowship from the Wellcome Trust (097152/Z/11/Z). A.W. is supported by an MRC Clinical Research Training Fellowship (MR/ N001524/1). D.L.B. is a Wellcome Senior Clinical Scientist (202747/Z/16/Z). A.B.S. was supported by an advanced Postdoc. Mobility Fellowship from the Swiss National Science Foundation (P00P3-158835) and National Institute for Health Research (NIHR) Oxford Biomedical Research Centre (BRC). M.V.H. works in a unit that receives funding from the UK Medical Research Council and is supported by a British Heart Foundation Intermediate Clinical Research Fellowship (FS/18/23/33512) and the NIHR Oxford Biomedical Research Centre. The PiNS study is supported by a strategic award from the Wellcome Trust to the Wellcome Pain Consortium (Ref. 102645). We thank the Oxford Genomics Centre at the Wellcome Centre for Human Genetics (funded by Wellcome Trust grant reference 203141/Z/16/Z) for the generation and initial processing of sequencing data. We also thank our clinical collaborators of the NDOMRS Hand Research Group for the collection of surgical specimens (https://www.ndorms.ox.ac.uk/research-groups/collaborative-hand-research-group). The Genotype-Tissue Expression (GTEx) Project was supported by the Common Fund of the Office of the Director of the National Institutes of Health and by NCI, NHGRI, NHLBI, NIDA, NIMH, and NINDS. The data used for the analyses described in this manuscript were obtained from the GTEx Portal on 8 February 2018.

## Author contributions

D.F. and D.L.B. designed and supervised the study. A.B.S. organised and carried out subject recruitment and phenotyping. A.W. and M.N. carried out the genetic association analysis and computational analyses. R.M. and K.K. performed the replication GWAS in the Estonian cohort. A.W., R.W.S. and M.V.H. performed the Mendelian randomisation study. A.W., M.N. and A.B.S. performed biological material collection, and G.B. carried out the analysis of the RNA-Seq data. A.W. and D.F. drafted the manuscript. All authors contributed to the final version of the paper.

## Additional information

**Competing interests:** The authors declare no competing interests.

