## [Peer Review File · Nature Communications]

Reviewer #1 (Remarks to the Author):

This paper reports a genome-wide association study for carpal tunnel syndrome (CTS), the commonest entrapment neuropathy. The authors undertook the genome-wide association study (GWAS), using 12,106 CTS cases and 387,347 controls from the UK Biobank. They discovered 13 novel genome-wide significant loci for CTS, and identified likely causal genes in these loci. They also adjusted the analysis for the presence of selected systemic conditions, such as diabetes, obesity, rheumatoid arthritis, and hypothyroidism (which were suggested to be among risk factors for CTS). Also, a finding that on average, CTS patients are shorter in height than controls, is interesting.

Importantly, using RNA sequencing from the surgically resected tenosynovium from CTS surgeries, the authors demonstrated differential expression of some of their identified genes. Expression quantitative trait loci (eQTL) were assessed for the candidate causal variants, however, not in the most relevant for CTS tissue. In general, since etiology of CTS involves entrapment of the median nerve at the level of the wrist, the suggestion that some of the identified genes, implicated in growth and extracellular matrix architecture contribute to the genetic predisposition to CTS, makes sense.

This is a straightforward GWAS, performed by an experienced group, with large numbers in the discovery cohort. The manuscript is clearly and consistently written. However, there are several outstanding issues:

Methods/ RESULTS: Participants with diagnostic codes for peripheral neuropathies other than CTS were excluded from both samples of cases and controls. This might be unwarranted, and these participants can be analyzed with the CTS for sensitivity's sake (see p. 10, the case of HNPP disorder and the PMP22 gene).

Using the weighted genetic risk score (wGRS) calculated from the same individuals who were included in the GWAS and applying to the discovery dataset: It would need an independent cohort to validate whether this wGRS is predictive of the disease's severity.

The authors conditioned on the disease status of diabetes, rheumatoid arthritis, hypothyroidism and obesity. It is unclear why Mendelian Randomization was not attempted instead. Why the conditional analysis did not include height as a covariate?

OTHER COMMENTS:

The cohorts used in this study were mostly composed of European-descent populations; a need to expand to multi-ethnic samples should be noted.

Clinical application of the knowledge this paper generated is not obvious.

p. 5: it is unclear whether "547,011 common-frequency SNPs" were imputed or genotyped.

It was shown that mutations in the gene SH3TC2, associated with Charcot-Marie-Tooth, confer susceptibility to neuropathy, including CTS. Did the authors try to test whether this gene is enriched in their analysis? (The same in regards to HNPP-associated PMP22 gene).

Similarly, there was a GWAS for Dupuytren Disease – which is a related Fibrosis and might share etiology with the CTS (Ng et al. Am J Hum Genet. 2017). The authors should have a look on the genetic correlation between these conditions.

Unclear, why Supplementary Table 1 does not include “obesity”.

Supplementary Figure 2: the very next after the “body height” among top enriched set of the SNPs was “membranous glomerulonephritis”; “kidney disease” was also enriched. Do the authors have any comments/speculations why? (On the other hand, “rheumatic disease” and skeletal/bone conditions did not make it to the top, - any idea why?)

Reviewer #2 (Remarks to the Author):

This is an extremely well written GWAS manuscript that reports several novel genome-wide significant signals at/close to compelling genes. The authors have a large CTS case cohort and a considerably larger control cohort, all derived from the UK Biobank.

All of the appropriate QC and analysis checks have been undertaken, whilst functional follow up has included public in-silico data as well as specific RNAseq data generated by the authors. The interpretation of the results is thoughtful and reasonable and is not at all exaggerated. There is one major issue that the authors freely acknowledge and do defend - the lack of replication. Replication is a mainstay of GWAS and its absence here means that some of the significant signals will represent type I error. Its not possible to predict how many.

Considering the mitigation provided and the current lack of genetic data for the disease and the potential insight provided by the signals, I'm minded to overlook this weakness.

I have no suggested changes to make regarding manuscript structure, presentation or clarity. It reads extremely well and has been performed to a very high level. Figures and tables are all clear and appropriate.

Reviewer #3 (Remarks to the Author):

Summary:

The manuscript submitted by Wiberg and colleagues describes the first ever genome-wide association study (GWAS) of Carpal Tunnel Syndrome [CTS, (N = 12,106 cases / 387,347 controls)]. Variants at 13 loci reached genome-wide significance ($P < 5 \times 10^{-8}$), however these associations were not followed up in an independent replication study. The authors attempted identify the most likely causal gene(s) underpinning each association signal by identifying missense coding variants, mining RegulomeDB and performing an eQTL study using GTEX expression data from several tissues. Three candidate genes were identified [i.e. ADAMTS17 and ADAMTS10 (missense coding variants) and EFEMP1 (regulomeDB and eQTL)] and a complementary RNA-seq study confirmed that all three genes were expressed in the tenosynovium of CTS patients.

General comment:

A brief search of the literature, suggests that the genetic architecture of CTS has not been thoroughly investigated using genome-wide methodologies. Consequently, Wiberg and colleagues are presented with an exciting opportunity to thoroughly investigate the genetic architecture of CTS. For example, computationally efficient tools such as genomic-relatedness-based restricted maximum-likelihood (Yang et al 2011, American Journal of Human Genetics), and LD Score regression (Bulik-Sullivan et al 2015, Nature Genetics) can be easily implemented to estimate the proportion of CTS risk (i.e. SNP heritability) explained by all genotyped/imputed genetic markers using unrelated individuals from the UK-Biobank Study, or summary results statistics from their CTS GWAS. Furthermore recent modifications to the LD Score Regression method enable one to partition the SNP heritability across functional genomic categories and tissue types, providing valuable insights into the genetic architecture and the molecular mechanisms underlying the regulation of gene expression relevant to CTS (Lui et al 2017, American Journal of Human Genetics). Moreover, LD Score regression, as implemented in LD-HUB (Zheng et al 2017, Bioinformatics), can also be used to establish whether the genetic architecture influencing CTS is shared with other traits and disease (i.e. obesity, diabetes and rheumatoid arthritis etc). Collectively, findings from these analyses could provide valuable insights into the genetic underpinnings of CTS and could be used inform future studies and identify putative risk factors of CTS that have a shared genetic basis, and that could in theory be targeted for future CTS intervention.

Unfortunately the study described by Wiberg and colleagues does not address these fundamental questions. I also have several major concerns with regards to the methods used in the study (described below). For these reasons I do not feel that the study is suitable for publication in the premier journal Nature Communications.

Major concerns:

1. The inclusion of disease status (namely: diabetes, rheumatoid arthritis, hypothyroidism and obesity) as covariates in the GWAS is not explicitly justified. I find this practice questionable as the inclusion of these heritable covariates could bias the effect estimates of variants that exert pleiotropic effects on CTS and these four disease phenotypes. Furthermore, the adjustment of heritable covariates may also induce spurious associations through collider bias (Aschard et al 2015, American Journal of Human Genetics). Similarly, the exclusion of subjects with peripheral

neuropathy could also induce a form of selection bias. It is generally good practice to always make the results from the minimally adjusted model available.

2. Correction for genomic inflation factor as estimated by λ_{GC} is considered to be overly conservative and could bias some downstream analyses. Consider using LD score regression to quantify the proportion of inflation due to polygenicity versus confounding before correcting for genomic inflation (Bulik-Sullivan et al 2015, Nature Genetics).

3. The authors state that the large sample size of their study mitigates the lack of a replication sample. I disagree wholeheartedly. Large sample sizes cannot rule out chance findings, and are unlikely to guard against artifacts that occur as a result of uncontrolled biases specific to one, but not a second independent replication sample.

4. The strategies used to identify causal genes underpinning each association signal could be improved substantially. Consider using more up to date methods such as Summary based Mendelian Randomization (SMR: Zhu et al 2016, Nature Genetics and Zeng et al 2018, Nature Communications).

5. The authors define a list of CTS genes for gene set enrichment analysis by selecting the closest gene to each sentinel CTS association. This approach could be improved by performing a genome-wide gene based test of association (e.g. MAGMA: de Leeuw et al 2015 PLOS Computational Biology) and running gene set enrichment analysis on the list of associated genes that meet the appropriate genome-wide significance threshold. This can be easily performed using FUMA (Watanabe et al 2017, Nature Communications).

Reviewers' comments:

Reviewer #1 (Remarks to the Author):

This paper reports a genome-wide association study for carpal tunnel syndrome (CTS), the commonest entrapment neuropathy. The authors undertook the genome-wide association study (GWAS), using 12,106 CTS cases and 387,347 controls from the UK Biobank. They discovered 13 novel genome-wide significant loci for CTS, and identified likely causal genes in these loci. They also adjusted the analysis for the presence of selected systemic conditions, such as diabetes, obesity, rheumatoid arthritis, and hypothyroidism (which were suggested to be among risk factors for CTS). Also, a finding that on average, CTS patients are shorter in height than controls, is interesting.

Importantly, using RNA sequencing from the surgically resected tenosynovium from CTS surgeries, the authors demonstrated differential expression of some of their identified genes. Expression quantitative trait loci (eQTL) were assessed for the candidate causal variants, however, not in the most relevant for CTS tissue. In general, since etiology of CTS involves entrapment of the median nerve at the level of the wrist, the suggestion that some of the identified genes, implicated in growth and extracellular matrix architecture contribute to the genetic predisposition to CTS, makes sense.

This is a straightforward GWAS, performed by an experienced group, with large numbers in the discovery cohort. The manuscript is clearly and consistently written. However, there are several outstanding issues:

Methods/ RESULTS: Participants with diagnostic codes for peripheral neuropathies other than CTS were excluded from both samples of cases and controls. This might be unwarranted, and these participants can be analyzed with the CTS for sensitivity's sake (see p. 10, the case of HNPP disorder and the PMP22 gene).

We have taken this point into consideration, and our main analysis has now been performed by not excluding participants with diagnostic codes for peripheral neuropathies.

Using the weighted genetic risk score (wGRS) calculated from the same individuals who were included in the GWAS and applying to the discovery dataset: It would need an independent cohort to validate whether this wGRS is predictive of the disease's severity.

We are grateful to the reviewer for highlighting this. Our wGRS is constructed from our GWAS summary results and applied to the same individuals, and therefore it is entirely to be expected that the cases will have a higher wGRS than the controls. As such, we do not make the circular claim that this in itself is a significant finding (we merely state the scores for cases vs controls in order to illustrate the absolute difference in wGRS between these two groups). So as to make this clearer, we have now prefaced the relevant sentence with, "As expected, wGRS in CTS cases... (page 12, line 2)".

What is interesting, and of potential relevance to the issue of lack of replication, however, is that this score is associated with CTS severity within our cohort. We would expect that CTS patients who have undergone an operation have a phenotypically more severe form of CTS compared to those CTS patients who have not. And we find that the those who have

undergone an operation for CTS have a significantly higher wGRS than the latter group. This provides us with some degree of internal validation of (1) the accuracy of our phenotyping methodology, and (2) the fact that the GWAS hits on which the wGRS is based are not spurious. In the absence of an independent cohort in which to test our wGRS, we believe it is fair to illustrate the differences between these two ‘sub-cohorts’ to show that wGRS appears to associate with disease severity; we do not make any claims beyond that.

The authors conditioned on the disease status of diabetes, rheumatoid arthritis, hypothyroidism and obesity. It is unclear why Mendelian Randomization was not attempted instead. Why the conditional analysis did not include height as a covariate?

Firstly, thank you for alerting us to the possibility of collider bias by conditioning on too many disease covariates (as explained in further detail by Reviewer #3). In this new analysis, we no longer condition on these disease covariates, and use only sex and genotyping platform as covariates in our GWAS analysis. The original reason for conditioning on these disease covariates was because they are known risk factors for CTS. We are not entirely clear why Reviewer #1 has suggested Mendelian randomisation (which establishes causality between an exposure and an outcome) as an alternative to conditioning in this context, as the rationale behind conditioning and MR are quite different.

Regarding the question of why the conditional analysis did not include height as a covariate, firstly, if this reviewer is suggesting that we should not condition on diseases associated with CTS such as diabetes and rheumatoid arthritis, we would think that the

same should apply for height. Secondly, conditioning for height in a CTS GWAS is somewhat akin to conditioning on BMI in a diabetes GWAS – and would tend to attenuate the genetic architecture of CTS that has arisen from pathways involving height. We also note that conditioning on anthropometric traits can lead to bias, as illustrated in a recent article (<https://www.ncbi.nlm.nih.gov/pubmed/29520038>).

In order to disentangle the intriguing relationship between height and CTS, we have taken the reviewer's suggestion of performing a Mendelian randomisation analysis, and we find compelling evidence that height is causally implicated in the aetiology of CTS.

OTHER COMMENTS:

The cohorts used in this study were mostly composed of European-descent populations; a need to expand to multi-ethnic samples should be noted.

We have now mentioned the need to extend the study to non-European ancestry populations in the Discussion (page 20, line 23).

Clinical application of the knowledge this paper generated is not obvious.

We do not feel that this our manuscript is particularly lacking in clinical applications compared to other GWAS papers, especially considering that this is the first ever GWAS performed in this disease. We illustrate that a genetic risk score correlates with CTS disease severity, and we provide new evidence of a causal relationship of height with CTS.

Furthermore, we also suggest miR-338-5p, which has been shown to inhibit the synthesis of EFEMP1 (a key gene implicated in our GWAS), to be a potential therapeutic target (page 18, line 13).

Finally, this new GWAS has uncovered three additional loci, and we have now implicated three genes in the TGF- β /Smad signaling pathway, a well-studied pathway in several diseases. We have dedicated a paragraph to discussing the relevance of these findings, and explain that the efficacy of corticosteroid injections into the carpal tunnel (a commonly performed procedure) could potentially be mediated through modulation of this pathway. (page 19, line 4)

p. 5: it is unclear whether “547,011 common-frequency SNPs” were imputed or genotyped.

The 547,011 SNPs were genotyped, and this has been made clear in the manuscript (page 25, line 11).

It was shown that mutations in the gene SH3TC2, associated with Charcot-Marie-Tooth, confer susceptibility to neuropathy, including CTS. Did the authors try to test whether this gene is enriched in their analysis? (The same in regards to HNPP-associated PMP22 gene). Similarly, there was a GWAS for Dupuytren Disease – which is a related Fibrosis and might share etiology with the CTS (Ng et al. Am J Hum Genet. 2017). The authors should have a look on the genetic correlation between these conditions.

Regarding the PMP22 and SH3TC2 genes, we scrutinised our GWAS summary statistics to look for possible enrichment for these neuronal genes. We examined all SNPs located

within +/- 1Mb of the genes (PMP22: 17:148982150-149063174; SH3TC2: 5:148982150-149063174), and the most significantly associated SNP at the PMP22 locus was rs71361436 ($p=2.7 \times 10^{-4}$), and rs17709363 ($p=9.7 \times 10^{-4}$) at the SH3TC2 locus. Therefore, there was minimal evidence for enrichment for these genes in the CTS GWAS, and we have dedicated a paragraph to this in the Discussion (page 19).

Regarding Dupuytren's Disease, the paper cited by the reviewer (Ng et al. Am J Hum Genet. 2017) came from our group. We are therefore familiar with the polygenic architecture of Dupuytren's Disease, and we politely disagree with the reviewer's statement that CTS is a "related fibrosis" to Dupuytren's Disease.

To substantiate this, we performed a cross-phenotype association study in CPASSOC (Li X & Zhu X, Methods Mol Biol 2017), using the summary data from our Dupuytren's Disease GWAS and the current CTS GWAS, to look for enrichment for the two traits across all genomic regions. Firstly, there were no SNPs that are genome-wide significant ($p < 5 \times 10^{-8}$) in all three of: the CTS GWAS, the Dupuytren's GWAS, the meta-analysed CPASSOC results.

We next examined SNPs that were genome-wide significant in the CTS GWAS and in the CPASSOC results, and found that none of these were remotely suggestive of association in the Dupuytren's GWAS (the minimum p-value was 0.0088 for rs4356642). Finally, we examined the set of SNPs that were genome-wide significant in the Dupuytren's GWAS and in the CPASSOC results – none of the SNPs were suggestive of significance at $p < 1 \times 10^{-5}$ in the CTS GWAS; three SNPs had very nominally "suggestive" p-values (ranging from $p=7.4 \times 10^{-5}$ to $p=9.8 \times 10^{-5}$). These three SNPs reside at the WNT7B locus on

chromosome 22 that we reported in the Dupuytren's paper. However, given the very lax definition of a "suggestive" p-value for these SNPs, we are strongly inclined to disregard them as evidence of enrichment across the two phenotypes. We therefore conclude that there is currently no evidence of a shared genetic architecture between CTS and Dupuytren's Disease. We do not feel that this negative finding warrants reporting in this manuscript, but would be happy to include it as supplementary data should you or the editor feel it necessary.

Unclear, why Supplementary Table 1 does not include "obesity".

We have no longer conditioned on "obesity" in our GWAS, and therefore, do not provide this information.

Supplementary Figure 2: the very next after the "body height" among top enriched set of the SNPs was "membranous glomerulonephritis"; "kidney disease" was also enriched. Do the authors have any comments/speculations why? (On the other hand, "rheumatic disease" and skeletal/bone conditions did not make it to the top, - any idea why?)

We have used the SNP-based enrichment methods proposed by Reviewer #3 (FUMA) - as such, we no longer provide the results of SNP-based enrichment using this method. As a general point, we are inherently sceptical of the value of the various computational tools available to map GWAS summary statistics onto disease ontologies in the context of CTS. The ontologies that exist in these computational tools are invariably subject to publication bias; whereas numerous GWAS have been conducted in e.g. autoimmune, cardiovascular and neuro-psychiatric diseases, our study is the first to investigate an entrapment neuropathy through genome-wide association. Therefore, we

feel that many of the disease (and other) ontologies that are mapped onto our GWAS summary statistics must be interpreted with caution.

Reviewer #2 (Remarks to the Author):

This is an extremely well written GWAS manuscript that reports several novel genome-wide significant signals at/close to compelling genes. The authors have a large CTS case cohort and a considerably larger control cohort, all derived from the UK Biobank.

All of the appropriate QC and analysis checks have been undertaken, whilst functional follow up has included public in-silico data as well as specific RNAseq data generated by the authors. The interpretation of the results is thoughtful and reasonable and is not at all exaggerated. There is one major issue that the authors freely acknowledge and do defend - the lack of replication. Replication is a mainstay of GWAS and its absence here means that some of the significant signals will represent type I error. Its not possible to predict how many.

Considering the mitigation provided and the current lack of genetic data for the disease and the potential insight provided by the signals, I'm minded to overlook this weakness.

I have no suggested changes to make regarding manuscript structure, presentation or clarity.

It reads extremely well and has been performed to a very high level. Figures and tables are all clear and appropriate.

We thank the reviewer for these positive comments. We are glad that the reviewer acknowledges that we are measured in our interpretation of our novel findings and that we are explicit about the issue of replication. In the interest of transparency we include further information about or attempts at replication after the response to Reviewer #3.

Reviewer #3 (Remarks to the Author):

Summary:

The manuscript submitted by Wiberg and colleagues describes the first ever genome-wide association study (GWAS) of Carpal Tunnel Syndrome [CTS, (N = 12,106 cases / 387,347 controls)]. Variants at 13 loci reached genome-wide significance ($P < 5 \times 10^{-8}$), however these associations were not followed up in an independent replication study. The authors attempted to identify the most likely causal gene(s) underpinning each association signal by identifying missense coding variants, mining RegulomeDB and performing an eQTL study using GTEX expression data from several tissues. Three candidate genes were identified [i.e. ADAMTS17 and ADAMTS10 (missense coding variants) and EFEMP1 (regulomeDB and eQTL)] and a complementary RNA-seq study confirmed that all three genes were expressed in the tenosynovium of CTS patients.

General comment:

A brief search of the literature, suggests that the genetic architecture of CTS has not been thoroughly investigated using genome-wide methodologies. Consequently, Wiberg and colleagues are presented with an exciting opportunity to thoroughly investigate the genetic architecture of CTS. For example, computationally efficient tools such as genomic-relatedness-based restricted maximum-likelihood (Yang et al 2011, American Journal of Human Genetics), and LD Score regression (Bulik-Sullivan et al 2015, Nature Genetics) can be easily implemented to estimate the proportion of CTS risk (i.e. SNP heritability) explained

by all genotyped/imputed genetic markers using unrelated individuals from the UK-Biobank Study, or summary results statistics from their CTS GWAS.

Furthermore recent modifications to the LD Score Regression method enable one to partition the SNP heritability across functional genomic categories and tissue types, providing valuable insights into the genetic architecture and the molecular mechanisms underlying the regulation of gene expression relevant to CTS (Lui et al 2017, American Journal of Human Genetics). Moreover, LD Score regression, as implemented in LD-HUB (Zheng et al 2017, Bioinformatics), can also be used to establish whether the genetic architecture influencing CTS is shared with other traits and disease (i.e. obesity, diabetes and rheumatoid arthritis etc). Collectively, findings from these analyses could provide valuable insights into the genetic underpinnings of CTS and could be used inform future studies and identify putative risk factors of CTS that have a shared genetic basis, and that could in theory be targeted for future CTS intervention.

As requested, we have now performed LD Score Regression to estimate SNP-based heritability, and have performed further analyses to partition the heritability across functional categories and also cell/tissue types. The latter analysis was particularly striking in that it suggested that osteoblasts are the cell type that shows the greatest enrichment for CTS heritability (along with several other musculoskeletal and connective tissues), consistent with our thesis of the importance of altered skeletal growth in the aetiology of CTS.

We have also used LD Score Regression to investigate genetic correlations between CTS and the traits mentioned above. We found statistically significant correlations with

anthropometric measures such as BMI and height, whereas other traits such as type 2 diabetes and rheumatoid arthritis did not meet the Bonferroni-corrected significance threshold. (Supplementary Table 7).

Unfortunately the study described by Wiberg and colleagues does not address these fundamental questions. I also have several major concerns with regards to the methods used in the study (described below). For these reasons I do not feel that the study is suitable for publication in the premier journal Nature Communications.

Major concerns:

1. The inclusion of disease status (namely: diabetes, rheumatoid arthritis, hypothyroidism and obesity) as covariates in the GWAS is not explicitly justified. I find this practice questionable as the inclusion of these heritable covariates could bias the effect estimates of variants that exert pleiotropic effects on CTS and these four disease phenotypes. Furthermore, the adjustment of heritable covariates may also induce spurious associations through collider bias (Aschard et al 2015, American Journal of Human Genetics). Similarly, the exclusion of subjects with peripheral neuropathy could also induce a form of selection bias. It is generally good practice to always make the results from the minimally adjusted model available.

As mentioned above in a reply to Reviewer #1, we have addressed this issue by (1) not excluding participants with a peripheral neuropathy diagnosis, and (2) not adjusting for the disease covariates. Thus, our GWAS now only conditions on GWAS platform and sex.

2. Correction for genomic inflation factor as estimated by λ_{GC} is considered to be overly conservative and could bias some downstream analyses. Consider using LD score regression

to quantify the proportion of inflation due to polygenicity versus confounding before correcting for genomic inflation (Bulik-Sullivan et al 2015, Nature Genetics).

We have now calculated the LD score regression intercept, with an associated attenuation ratio (intercept = 1.015; attenuation ratio = 0.073), consistent with polygenicity and the large sample size.

3. The authors state that the large sample size of their study mitigates the lack of a replication sample. I disagree wholeheartedly. Large sample sizes cannot rule out chance findings, and are unlikely to guard against artifacts that occur as a result of uncontrolled biases specific to one, but not a second independent replication sample.

We concede that there is the possibility of false positive associations in a single-stage GWAS, and further acknowledge the need to replicate the findings in a large independent cohort (page 20, line 19).

4. The strategies used to identify causal genes underpinning each association signal could be improved substantially. Consider using more up to date methods such as Summary based Mendelian Randomization (SMR: Zhu et al 2016, Nature Genetics and Zeng et al 2018, Nature Communications).

We have used 25 genes mapped to our loci by FUMA and 17 genes mapped by MAGMA gene-based association, rather than the genes in closest proximity.

Summary-based Mendelian Randomisation (SMR) requires eQTL data, of which there are publicly available summary data for various tissue types. These include the 53 GTEx tissue types, and various eQTLs in whole blood. One obstacle that we come across time and again when performing these types of computational analyses on our CTS data is that CTS (and other entrapment neuropathies) have been hitherto relatively underinvestigated. As such, the tissue-specific data for these conditions simply do not exist.

The findings from our study strongly suggest that the tissues of interest (where CTS-predisposing risk genes are principally likely to act) are: (1) the tenosynovium within the carpal tunnel, and (2) bone (causing altered growth and skeletal proportions of the upper limb). It is notable that there are no GTEx eQTLs for synovium or bone. We have, as a result, resorted to using the GTEx eQTL data for “transformed fibroblasts”, which are a constituent of tenosynovium; however, it goes without saying that transformed fibroblasts will be phenotypically different from the fibroblasts that are found in carpal tunnel tenosynovium in vivo.

5. The authors define a list of CTS genes for gene set enrichment analysis by selecting the closest gene to each sentinel CTS association. This approach could be improved by performing a genome-wide gene based test of association (e.g. MAGMA: de Leeuw et al 2015 PLOS Computational Biology) and running gene set enrichment analysis on the list of associated genes that meet the appropriate genome-wide significance threshold. This can be easily performed using FUMA (Watanabe et al 2017, Nature Communications).

We have now performed various analyses in FUMA, including MAGMA, and the Results and Discussion sections have been considerably expanded accordingly.

As a final comment, we are thankful to all the reviewers for their insightful comments and suggestions, and we agree that the paper is substantially improved by the new analyses that have been undertaken.

Replication in Estonian Cohort

Further to your decision regarding this manuscript on 5th of June 2018 and the issue of replication that was raised, we have made substantial efforts to try to replicate this GWAS by reaching out to several international biobanks. The vast majority of these potential collaborators either did not collect CTS data, or had very small cohorts (of no more than a few hundred cases). However, the Estonian Institute of Genomics at the University of Tartu (EGCUT) had a reasonably-sized cohort of CTS cases, and we therefore initiated a collaboration with Dr Reedik Mägi and his team to establish whether we could replicate our GWAS in EGCUT.

The EGCUT cohort identified 4,087 cases and 43,750 controls (disease prevalence 8.5% overall – 4.5% in men and 11.0% in women). Diagnoses were made using a mixture of questionnaire data (self-diagnosis) and diagnoses by a medical professional (although neither ICD-10 nor OPCS codes were used). Association analysis was performed using year of birth, sex and the first 10 principal components as covariates.

We initially scrutinised p-values in the EGCUT cohort for the 16 sentinel SNPs at the associated loci from our UK Biobank GWAS. Results are summarised in the table below. Of the 16 loci, one locus (rs62621197, ADAMTS10 missense) replicates at a Bonferroni-corrected significance threshold of $p < 0.0031$. Furthermore, the odds ratios at 13 of 16 loci are directionally concordant with our data. Thus, while these results are not all together unpromising, we do have several serious concerns:

1. CTS prevalence and age of onset:

One striking observation was that the prevalence of CTS in the EGCUT cohort is substantially higher than in UK Biobank (8.5% vs 3.1%). Moreover, the age distribution of CTS cases in EGCUT (shown in the graph, below) appears normally distributed around a mean of ~52 years. This is highly surprising to us as clinicians who treat CTS, as CTS is very rare under the age of 40 (apart from pregnancy-related CTS, which has a different aetiology). UK Biobank participants were recruited at ages 40-69 (the peak age for CTS incidence), and the prevalence was still under 5% (the CTS cases in our cohort, incidentally, have a mean age of 69).

In UK Biobank, we have performed a validation study of CTS diagnoses by scrutinising hundreds of medical records of putative CTS cases, and we have determined that the positive predictive value of a CTS 'case' being a true case is 94%. In addition, 94% of our CTS cases have at least one operation code, suggesting that our cohort are, on the whole, on the phenotypically more severe end of the spectrum. Finally, 80% of our CTS cases have at least one ICD10 or OPCS code for CTS, meaning that our reliance on self-reported questionnaire diagnoses is very low. On the basis of the above, we are highly confident in the quality of the phenotyping for CTS cases within UK Biobank.

We therefore wonder whether the unusually high prevalence of CTS in EGCUT may be in large part due to misclassification bias. The high female preponderance in EGCUT (F:M = 5:1) and the surprisingly young age at diagnosis suggest that the EGCUT cohort may include a significant number of CTS cases during pregnancy – a common and usually transient phenomenon due to fluid retention in pregnancy, which is quite distinct in terms of pathophysiology from idiopathic CTS. As such, the genetic architecture of CTS during

pregnancy is likely different to that in the general population, which could account for these differences. Moreover, there may be further misclassification based on over-reliance on self-reported questionnaire data for diagnosis.

2. Lack of power:

Our other major concern regarding the EGCUT data pertains to lack of statistical power. Despite having >4,000 cases, this still only represents approximately one-third of the cases included in the UK Biobank discovery cohort. With controls included, the EGCUT cohort is only 12% of the size of the UK Biobank cohort. Based on the number of Estonian cases and controls and the allelic odds ratios from our GWAS, we calculated the power to detect an association at the Bonferroni-corrected p-value threshold of $p=0.0031$, assuming a population disease prevalence of 5%. As is apparent from the results shown in the right-hand section of the table below, under “Power (4087 cases)”, we are significantly underpowered at the majority of loci; only at 4/16 loci do we have > 80% power to detect an association at this threshold, and that is assuming that the majority of the 4087 cases are true cases. When we repeat the power calculation with a more conservative 2000 cases (yielding a more realistic case fraction of 4.2% - still higher than in UK Biobank), the power is substantially diminished at all loci (“Power (2000 cases)”, far right column).

Concluding remarks:

In summary, we have attempted replication of our findings in the best cohort that was available to us. Despite the limitations of the replication cohort discussed above, our association p-values would become more significant at 13/16 loci if we were to formally

meta-analyse the two GWAS. However, we would be doing so with considerable reservations, and our Estonian collaborators share our concerns regarding the unexpectedly high case fraction of CTS, and agree that we are underpowered to replicate our associations in this cohort.

We feel that we have gone to great lengths to ensure that our UK Biobank CTS cases have been robustly phenotyped, and in the current version of the manuscript, we devote a substantial part of the Discussion to outlining the strengths and weaknesses of the study, including the risk of false positive discoveries in a single-stage GWAS and the need for replication. However, for the reasons outlined above, we do not feel that this Estonian CTS cohort is sufficiently robust for these purposes.

We present these data to you in the interests of full transparency, and kindly ask that you exercise your editorial judgment in deciding whether our attempt at replication ought to be included in our manuscript (with caveats and limitations). We thank you again for taking the time to re-review our manuscript.

	UK BIOBANK					ESTONIA							
rsID	Effect Allele	EAF Cases	EAF Controls	OR (95% CI)	P-value	Effect Allele	EAF Cases	EAF Controls	(95% CI)	P-value	Directionally concordant?	Power (4087 cases)	Power (2000 cases)
rs12406439	T	0.605	0.587	1.08 (1.05-1.11)	1.10x10-8	T	0.627245	0.617024	1.04 (0.99-1.09)	0.0503565	Yes	0.61	0.26
rs12104955	C	0.517	0.499	1.07 (1.05-1.10)	3.90x10-8	C	0.455346	0.446615	1.03 (0.99-1.08)	0.116235	Yes	0.51	0.2
rs3791679	G	0.244	0.225	1.11 (1.08-1.15)	2.00x10-12	G	0.276242	0.270949	1.02 (0.98-1.08)	0.327234	Yes	0.88	0.49
rs1025128	C	0.582	0.563	1.08 (1.05-1.11)	2.80x10-9	C	0.676057	0.668149	1.03 (0.99-1.09)	0.105171	Yes	0.63	0.24
rs847139	C	0.805	0.787	1.11 (1.07-1.14)	7.20x10-11	C	0.819721	0.815383	1.03 (0.97-1.09)	0.16907	Yes	0.56	0.29
rs1863190	T	0.78	0.76	1.12 (1.08-1.15)	5.40x10-13	T	0.707689	0.701424	1.03 (0.98-1.08)	0.264197	Yes	0.92	0.57
rs4678145	G	0.88	0.868	1.12 (1.08-1.16)	4.10x10-9	G	0.9135533	0.9054511	1.10 (1.02-1.20)	0.0137248	Yes	0.39	0.15
rs6843953	T	0.154	0.138	1.14 (1.10-1.18)	5.80x10-12	T	0.0960464	0.0923774	1.04 (0.97-1.13)	0.338336	Yes	0.67	0.3
rs3828889	C	0.748	0.732	1.09 (1.06-1.12)	1.70x10-8	C	0.692476	0.694025	0.99 (0.95-1.04)	0.757172	No	0.68	0.3
rs62422907	G	0.899	0.887	1.13 (1.09-1.18)	2.20x10-9	G	0.9111696	0.9163058	0.94 (0.86-1.01)	0.0877564	No	0.5	0.18
rs55841377	C	0.789	0.773	1.09 (1.06-1.13)	8.40x10-9	C	0.840796	0.838772	1.02 (0.95-1.08)	0.499781	Yes	0.39	0.15
rs6977081	G	0.685	0.668	1.08 (1.05-1.11)	1.20x10-8	G	0.610221	0.603373	1.03 (0.98-1.08)	0.238732	Yes	0.63	0.27
rs72725608	C	0.051	0.044	1.20 (1.13-1.27)	1.10x10-8	C	0.0334577	0.0307975	1.09 (0.96-1.24)	0.164652	Yes	0.45	0.18
rs1866745	A	0.369	0.35	1.09 (1.06-1.12)	4.20x10-10	A	0.324299	0.325032	0.997 (0.95-1.24)	0.847656	No	0.73	0.34
rs72755233	A	0.128	0.112	1.18 (1.13-1.22)	2.30x10-15	A	0.144951	0.138822	1.05 (0.99-1.12)	0.0855613	Yes	0.99	0.76
rs62621197	T	0.045	0.036	1.31 (1.22-1.40)	7.50x10-14	T	0.0488515	0.0415869	1.18 (1.06-1.31)	0.00209897	Yes	0.97	0.68

Reviewer #1 (Remarks to the Author):

The authors have provided appropriate responses to all the concerns this reviewer raised. There is one outstanding question though:

Previous comment by reviewer 1: "Participants with diagnostic codes for peripheral neuropathies other than CTS were excluded ..." (See also Rev. 2)

The authors have addressed this issue by (1) not excluding participants with a peripheral neuropathy diagnosis, and (2) not adjusting for other covariates. Thus, it is expected that Table 1 should be changed a bit, which hadn't happen. Pls. comment.

Reviewer #2 (Remarks to the Author):

Thank you for the responses to reviewer's comments. These were detailed and thoughtful.

Reviewer #3 (Remarks to the Author):

Reviewer #3, residual concerns:

1. Author:

As mentioned above in a reply to Reviewer #1, we have addressed this issue by (1) not excluding participants with a peripheral neuropathy diagnosis, and (2) not adjusting for the disease covariates. Thus, our GWAS now only conditions on GWAS platform and sex.

Reviewer: Could the authors please comment on why "year of birth" was included as a covariate in the regression model for replication, but not for discovery?

2. Author:

In order to disentangle the intriguing relationship between height and CTS, we have taken the reviewer's suggestion of performing a Mendelian randomisation analysis, and we find compelling evidence that height is causally implicated in the aetiology of CTS.

Reviewer:

Could the authors perform the bidirectional MR analysis, to rule out a possible causal effect of CTS on height?

3. Author:

Summary-based Mendelian Randomisation (SMR) requires eQTL data, of which there are publicly available summary data for various tissue types. These include the 53 GTEx tissue types, and various eQTLs in whole blood. One obstacle that we come across time and again when performing these types of computational analyses on our CTS data is that CTS (and other entrapment neuropathies) have been hitherto relatively under investigated. As such, the tissue-specific data for these conditions simply do not exist. The findings from our study strongly suggest that the tissues of interest (where CTS- predisposing risk genes are principally likely to act) are: (1) the tenosynovium within the carpal tunnel, and (2) bone (causing altered growth and skeletal proportions of the upper limb). It is notable that there are no GTEx eQTLs for synovium or bone. We have, as a result, resorted to using the GTEx eQTL data for "transformed fibroblasts", which are a constituent of tenosynovium; however, it goes without saying that transformed fibroblasts will be phenotypically different from the fibroblasts that are found in carpal tunnel tenosynovium in vivo. Summary-based Mendelian Randomisation (SMR) requires eQTL data, of which there are publicly available summary data for various tissue types. These include the 53 GTEx tissue types, and various eQTLs in whole blood. One obstacle that we come across time and again when performing these types of computational analyses on our CTS data is that CTS (and other entrapment neuropathies) have been hitherto relatively underinvestigated. As such, the tissue-specific data for these conditions simply do not exist.

Reviewer:

The authors investigate transformed fibroblasts with SMR, despite the gene-property analysis strongly implicating the tibial artery and coronary artery as tissues of interest (Suppl Figure 4). Although the link between these tissue types and CTS is not intuitive or immediately clear, it may be worthwhile to use SMR to potentially identify the plausibly causal gene(s) that contribute to these associations as they could implicate tangible disease pathways.

4. Author:

Replication in Estonian Cohort - Further to your decision regarding this manuscript on 5th of June 2018 and the issue of replication that was raised, we have made substantial efforts to try to replicate this GWAS by reaching out to several international biobanks. The vast majority of these potential collaborators either did not collect CTS data, or had very small cohorts (of no more than a few hundred cases). However, the Estonian Institute of Genomics at the University of Tartu (EGCUT) had

a reasonably-sized cohort of CTS cases, and we therefore initiated a collaboration with Dr Reedik Mägi and his team to establish whether we could replicate our GWAS in EGCUT.

Reviewer:

In the interest of transparency, I feel strongly that the replication attempt be included in the manuscript together with the caveats and limitations. Furthermore, I have a few comments in respect to the replication effort:

1. The UK-Biobank Study does not appear to be representative of the UK population as discussed in the following paper (PMID:29040562). I would caution against relying on population based estimates of disease prevalence as they appear to suffer from ascertainment bias.
2. Please provide descriptive summary statistics relating to key variables from both cohorts (e.g. height, weight, age etc) so that the comparability of the discovery and replication cohorts can be evaluated.
3. Please perform LD score regression to evaluate the genetic similarity between CTS as defined in Biobank to that defined in EGCUT using GWAS summary statistics. This may help to evaluate the comparability of disease definitions, assuming the replication sample is sufficiently powered.
4. To complement point 3, it would be beneficial to create a genetic risk score in the EGCUT study (using the 16 GWAS associated CTS SNPs and their weights) and test for an association with CTS in EGCUT.
5. Arguably, the Bonferonni corrected threshold of association may be too conservative, given that the attempt at replication is informed by the discovery GWAS. Could the authors supplement the table with power estimates using an alpha of 0.05, correcting for winner's curse if possible.
5. Since the release of the UK-Biobank Study, it is increasingly difficult to identify replication cohorts of sufficient sample size for robust replication. To partially address this issue, Yengo and colleagues (PMID:30124842) quantify replicability of their GWAS associations "jointly" by estimating the regression slope of SNP effect size estimated in the replication sample onto the SNP effect sizes (corrected for winner's curse effects) from their discovery. Perhaps the authors could attempt to evaluate their findings using this method, keeping in mind that it may not perform as well given that the current study identified 16 SNPs and not thousands as per the cited GIANT study.

Reviewers' comments:

Reviewer #1 (Remarks to the Author):

The authors have provided appropriate responses to all the concerns this reviewer raised. There is one outstanding question though:

Previous comment by reviewer 1: "Participants with diagnostic codes for peripheral neuropathies other than CTS were excluded ..." (See also Rev. 2)

The authors have addressed this issue by (1) not excluding participants with a peripheral neuropathy diagnosis, and (2) not adjusting for other covariates. Thus, it is expected that Table 1 should be changed a bit, which hadn't happen. Pls. comment.

If the reviewer is referring to Table 1 in the main text, it is substantially different from Table 1 in our original submission – there are 16 genome-wide significant SNPs, not 13, and the p-values/odds ratios have all changed due to the different methods used in our association study. The confusion could have arisen because tracked changes were turned off for this table, as the markings got very congested and the table became impossible to visualise. We apologise for not clarifying this at the time of resubmission.

If the reviewer is referring to Supplementary Table 1, we removed the diagnostic codes for all other diseases apart from CTS between our first and second submissions. The number of individuals with each of the diagnostic codes remained the same between our two previous manuscripts, as this table listed the numbers of individuals prior to QC. We realise that this is potentially confusing, so we have now modified the table to only display the number of individuals with each diagnostic code following sample QC.

Reviewer #2 (Remarks to the Author):

Thank you for the responses to reviewer's comments. These were detailed and thoughtful.

We thank the reviewer for their comments and their re-review of our manuscript.

Reviewer #3 (Remarks to the Author):

Reviewer #3, residual concerns:

1. Author:

As mentioned above in a reply to Reviewer #1, we have addressed this issue by (1) not excluding participants with a peripheral neuropathy diagnosis, and (2) not adjusting for the disease covariates. Thus, our GWAS now only conditions on GWAS platform and sex.

Reviewer: Could the authors please comment on why "year of birth" was included as a covariate in the regression model for replication, but not for discovery?

The Supplementary Materials now contains an entire section dedicated to our attempts at replication in the Estonian cohort. One of the tables demonstrates the demographics of the

UK Biobank vs EGCUT cohorts. The age difference in UK Biobank between cases and controls was a mere 2 years, and we therefore did not condition on age, especially in light of the issues that were raised by both reviewers 1 and 3 previously regarding conditioning on too many variables in our original discovery GWAS.

In contrast, in EGCUT, the age difference between cases and controls was 6 years in females and 7 years in males, hence the inclusion of age as a covariate in the replication GWAS. The EGCUT dataset is considerably different to the UK Biobank dataset, and our two groups have consequently developed different GWAS pipelines to reflect this.

2. Author:

In order to disentangle the intriguing relationship between height and CTS, we have taken the reviewer's suggestion of performing a Mendelian randomisation analysis, and we find compelling evidence that height is causally implicated in the aetiology of CTS.

Reviewer:

Could the authors perform the bidirectional MR analysis, to rule out a possible causal effect of CTS on height?

We think it is highly unlikely that a condition characterised by compression of a nerve in the hand in adulthood, well after the age of skeletal maturity, could have an effect on the height of a patient. However, we performed the analysis as requested, and the results are shown below.

The analysis used 13 of 16 SNPs as instrumental variables (three of our associated CTS SNPs did not appear in, and did not have a corresponding SNP in linkage disequilibrium within the GIANT height dataset).

	Method	Estimate	SE	95% CI (lower)	95% CI (upper)	P-value
Main Analysis	IVW	-0.122	0.063	-0.244	0.001	0.052
	MR-Egger	0.327	0.315	-0.290	0.944	0.299
	Intercept	-0.041	0.028	-0.095	0.014	0.147

The IVW result demonstrates very weak evidence for reverse causality, whereas the MR-Egger does not. On balance, this suggests that the direction of causality is almost certainly and exclusively from height to CTS, rather than vice versa. Given the biological implausibility of reverse causality in this context, we do not feel that these results ought to be included in our paper.

3. Author:

Summary-based Mendelian Randomisation (SMR) requires eQTL data, of which there are publicly available summary data for various tissue types. These include the 53 GTEx tissue types, and various eQTLs in whole blood. One obstacle that we come across time and again when performing these types of computational analyses on our CTS data is that CTS (and other entrapment neuropathies) have been hitherto relatively under investigated. As such, the tissue-specific data for these conditions simply do not exist. The findings from our study strongly suggest that the tissues of interest (where CTS- predisposing risk genes are principally likely to act) are: (1) the tenosynovium within the carpal tunnel, and (2) bone (causing altered growth and skeletal proportions of the upper limb). It is notable that there are no GTEx eQTLs for synovium or bone. We have, as a result, resorted to using the GTEx eQTL data for “transformed fibroblasts”, which are a constituent of tenosynovium; however, it

goes without saying that transformed fibroblasts will be phenotypically different from the fibroblasts that are found in carpal tunnel tenosynovium in vivo. Summary-based Mendelian Randomisation (SMR) requires eQTL data, of which there are publicly available summary data for various tissue types. These include the 53 GTEx tissue types, and various eQTLs in whole blood. One obstacle that we come across time and again when performing these types of computational analyses on our CTS data is that CTS (and other entrapment neuropathies) have been hitherto relatively underinvestigated. As such, the tissue-specific data for these conditions simply do not exist.

Reviewer:

The authors investigate transformed fibroblasts with SMR, despite the gene-property analysis strongly implicating the tibial artery and coronary artery as tissues of interest (Suppl Figure 4). Although the link between these tissue types and CTS is not intuitive or immediately clear, it may be worthwhile to use SMR to potentially identify the plausibly causal gene(s) that contribute to these associations as they could implicate tangible disease pathways.

As the reviewer acknowledges, the link between coronary artery, tibial artery, and carpal tunnel is highly tenuous from a physiological point of view. As we have explained, any association is likely caused by publication bias inherent in the databases selected for analysis. As such, we think that any analysis performed using these tissue types is likely to lead to false conclusions that would distract from the manuscript.

We had actually previously performed SMR for tibial artery and coronary artery, and did not find any additional genes implicated, and therefore did not include this in the manuscript.

4. Author:

Replication in Estonian Cohort - Further to your decision regarding this manuscript on 5th of June 2018 and the issue of replication that was raised, we have made substantial efforts to try to replicate this GWAS by reaching out to several international biobanks. The vast majority of these potential collaborators either did not collect CTS data, or had very small cohorts (of no more than a few hundred

cases). However, the Estonian Institute of Genomics at the University of Tartu (EGCUT) had a reasonably-sized cohort of CTS cases, and we therefore initiated a collaboration with Dr Reedik Mägi and his team to establish whether we could replicate our GWAS in EGCUT.

Reviewer:

In the interest of transparency, I feel strongly that the replication attempt be included in the manuscript together with the caveats and limitations. Furthermore, I have a few comments in respect to the replication effort:

As we stated in our reply, we are willing to publish this data, and the editor has suggested it is included in the supplementary materials.

1. The UK-Biobank Study does not appear to be representative of the UK population as discussed in the following paper (PMID:29040562). I would caution against relying on population based estimates of disease prevalence as they appear to suffer from ascertainment bias.

We agree that biobanks in general are not representative of the whole population, and this applies to all biobanks, not just UK Biobank. As we point out it is likely that the UK Biobank has a higher rate of CTS than the UK population because of the age of recruitment.

In fact, in our unpublished epidemiological study of CTS using National Health Service data for a 20-year period in England, we have found a primary incidence rate of carpal tunnel decompression surgery of 1.10 per 1000 person years (95% CI 1.02 to 1.17). The median (IQR) age at surgery was 57 (46.9 to 70.7 years), and 68% of this cohort were female. This gives us a very crude population prevalence of CTS requiring surgery of $665,090/54,000,000 = 1.2\%$.

Therefore, the UK Biobank population is likely to be representative of severe CTS cases in the population.

2. Please provide descriptive summary statistics relating to key variables from both cohorts (e.g. height, weight, age etc) so that the comparability of the discovery and replication cohorts can be evaluated.

We have included these descriptive summary statistics in the Supplementary Materials. A key finding was the age discrepancy between the two cohorts and the higher female preponderance in the replication cohort.

3. Please perform LD score regression to evaluate the genetic similarity between CTS as defined in Biobank to that defined in EGCUT using GWAS summary statistics. This may help to evaluate the comparability of disease definitions, assuming the replication sample is sufficiently powered.

We have performed this analysis, and the results are in the Supplementary Materials.

4. To complement point 3, it would be beneficial to create a genetic risk score in the EGCUT study (using the 16 GWAS associated CTS SNPs and their weights) and test for an association with CTS in EGCUT.

We have performed this analysis, and the results are in the Supplementary Materials.

5. Arguably, the Bonferroni corrected threshold of association may be too conservative, given that the attempt at replication is informed by the discovery GWAS. Could the authors supplement the table with power estimates using an alpha of 0.05, correcting for winner's curse if possible.

As phrased eloquently by S. Greenland (PMID: 22391267):

“Power refers only to future studies done on populations that look exactly like our sample with respect to the estimates from the sample used in the power calculation; for a study as completed (observed), it is analogous to giving odds on a horse race after seeing the outcome”.

We therefore respectfully question the value of repeating the power calculations with the new suggested parameters. We set our alpha level at a Bonferroni-corrected significance threshold, in line with all GWAS replication.

5. Since the release of the UK-Biobank Study, it is increasingly difficult to identify replication cohorts of sufficient sample size for robust replication. To partially address this issue, Yengo and colleagues (PMID:30124842) quantify replicability of their GWAS associations “jointly” by estimating the regression slope of SNP effect size estimated in the replication sample onto the SNP effect sizes (corrected for winner's curse effects) from their discovery. Perhaps the authors could attempt to evaluate their findings using this method, keeping in mind that it may not perform as well given that the current study identified 16 SNPs and not thousands as per the cited GIANT study.

We thank the reviewer for pointing out this study, which was published after submission of our article. The field continues to move quickly, making new and further analyses possible on a monthly basis. We have plotted the scatter graph and calculated the regression coefficient. However, as the reviewer points out, with only 16 SNPs, the regression slope calculation will lack precision, and we feel that this analysis is of questionable value and feel disinclined to include it in the Supplementary Materials.

$r^2 = 0.69$, F statistic: 31.3, $P < 0.0001$.

Reviewer #1 (Remarks to the Author):

I believe the authors responded to the critique satisfactorily. The paper is solid and well-rounded. However, one major drop-back had become obvious: an underdeveloped story of the carpal tunnel pathology and bone (size, density) and cartilage. There are the following findings: (a) in Suppl. Table 6 – top enriched cell types were Osteoblasts ($p=0.0005$) & Chondrocytes too (0.0139), and (b) candidate gene EFEMP1 is implicated in bone and cartilage. Therefore, this begs adding Suppl. Table 7 with genetic correlation between CTS and bone (and even osteoarthritis) phenotypes.

The authors partly correct that there are no GTEx eQTLs for bone, - but for Osteoblasts they exist.

Minor note: "Serum Urate overweight" (Suppl. Table 7) – is this correct?

Reviewer #3 (Remarks to the Author):

The authors have provided appropriate responses to most of my concerns, however given that the genetic correlation between the discovery and replication traits are similar, but not identical, I am not convinced that lack of statistical power is the main reason for lack of replication. The authors' unwillingness to perform a revised power calculation is also unfortunate, and their reasoning for not doing so is circular given that they use a post hoc power-calculation to justify lack of replication in the first place. I do however agree that a line needs to be drawn in regards to performing additional analysis, and therefore only request that the results of the replication attempt be clearly articulated in the main text of the manuscript. The readership should be made aware of these findings so that they can use their discretion when using the summary association results for downstream applications. The authors only refer to the replication attempt on the penultimate page of the discussion, using a single sentence, and consequently these results can be easily overlooked.

Reviewers' comments:

Reviewer #1 (Remarks to the Author):

I believe the authors responded the critique satisfactorily. The paper is solid and well-rounded.

We thank the reviewer for these positive comments, and for reviewing our paper again.

However, one major drop-back had become obvious: an underdeveloped story of the carpal tunnel pathology and bone (size, density) and cartilage. There are the fol. findings: (a) in Suppl. Table 6 – top enriched cell types were Osteoblasts ($p=0.0005$) & Chondrocytes too (0.0139), and (b) candidate gene EFEMP1 is implicated in bone and cartilage. Therefore, this begs adding Suppl. Table 7 with genetic correlation between CTS and bone (and even osteoarthritis) phenotypes.

As requested, we have added to our LD correlation analysis the only bone-related trait available on LD Hub – “bone mineral density”. We do indeed find a statistically significant correlation between CTS and lumbar spine bone mineral density, and we speculate on its significance (and the significance for height and osteoblast enrichment) in the context of EFEMP1 in the Discussion (page 18).

Supplementary Table 7 has been updated, accordingly.

The authors partly correct that there are no GTEx eQTLs for bone, - but for Osteoblasts they exist.

There are no GTEx eQTLs for osteoblasts. Osteoblast eQTLs do exist within the Franke Lab gene expression datasets, and we have acknowledged this in Supplementary Table 6 (where osteoblasts show the top enrichment out of all cell and tissue types).

Minor note: " Serum Urate overweight" (Supp. Table 7) – is this correct?

This is how the urate data are presented in LD Hub. The genome-wide data that LD Hub uses from this analysis is derived from Huffman et al, 2015: “Modulation of Genetic Associations with Serum Urate Levels by Body-Mass-Index in Humans”. The genome-wide significant loci were uncovered in overweight/obese individuals, hence the nomenclature. We therefore do not feel that the descriptor of the phenotype should be changed.

Reviewer #3 (Remarks to the Author):

The authors have provided appropriate responses to most of my concerns, however given that the genetic correlation between the discovery and replication traits are similar, but not identical, I am not convinced that lack of statistical power is the main reason for lack of replication. The authors' unwillingness to perform a revised power calculation is also unfortunate, and their reasoning for not doing so is circular given that they use a post hoc power-calculation justify lack of replication in the first place. I do however agree that a line needs to drawn in regards to performing additional analysis, and therefore only request that the results of the replication attempt be clearly articulated in the main text of the manuscript. The readership should be made aware of these findings so that they can use their discretion when using the summary association results for downstream applications. The authors only refer to the replication attempt on the penultimate page of the discussion, using a single sentence, and consequently these results can be easily overlooked.

We have addressed the concerns of this reviewer by expanding significantly on the part of the Discussion that raises the attempt at replication in an independent cohort, and comply with the Editor's previous suggestion that the replication study should be published in the Supplementary Materials:

“Our attempt to perform a replication in an underpowered independent cohort with a less stringent case definition are documented in the Supplementary Materials – in spite of the various limitations, the ADAMTS10 locus replicated at a Bonferroni-corrected significance threshold, and 13/16 loci showed directional concordance between the two GWAS, with a L.D. score regression-computed genetic correlation of 0.90.”